# One-for-All Model Initialization with Frequency-Domain Knowledge

## Abstract

Transferring knowledge by fine-tuning large-scale pre-trained networks has become a standard paradigm for downstream tasks, yet the knowledge of a pre-trained model is tightly coupled with monolithic architecture, which restricts flexible reuse across models of varying scales. In response to this challenge, recent approaches typically resort to either parameter selection, which fails to capture the interdependent structure of this knowledge, or parameter prediction using generative models that depend on impractical access to large network collections. In this paper, we empirically demonstrate that a model's foundational, task-agnostic knowledge – its "learngene" – is encoded within the low-frequency components of its weights, and can be efficiently inherited by downstream models. Based on this insight, we propose FRONT (FRequency dOmain kNowledge Transfer), a novel framework that uses the Discrete Cosine Transform (DCT) to isolate the low-frequency "learngene". This learngene can be seamlessly adapted to initialize models of arbitrary size via simple truncation or padding, a process that is entirely training-free. For enhanced performance, we propose an optional low-cost refinement process that introduces a spectral regularizer to further improve the learngene's transferability. Extensive experiments demonstrate that FRONT achieves the state-of-the-art performance, accelerates convergence by up to $15\times$ in vision tasks, and reduces training FLOPs by an average of 40.5% in language tasks.

## 1 Introduction

The paradigm of fine-tuning large-scale pre-trained models has become the de facto standard in deep learning, since it can transfer general-purpose knowledge learned from broad datasets to specific downstream tasks, consistently outperforming training from scratch (Kolesnikov et al., 2020; Hu et al., 2022). Although many powerful pre-trained models are now available from communities (Wolf et al., 2019; Wightman et al., 2019), their knowledge is tightly coupled with monolithic, computationally expensive architectures. This coupling creates significant challenges for adapting the foundational knowledge *from a single model to target models of varying scales*.

In response to this challenge, an emerging line of work attempts to predict parameters by training generative models on the weight distributions of model zoos (Knyazev et al., 2023; Wang et al., 2024a; Soro et al., 2024; Schürholt et al., 2024). While these methods can flexibly generate new initializations by sampling from a learned latent space, they typically require access to large, often homogeneous, collections of well-trained models, which is infeasible for large-scale networks. Furthermore, high computational overhead often restricts them to generating only a small fraction of the target parameters (e.g., normalization layers), leaving the majority randomly initialized and leading to suboptimal performance. A second line of work attempts to obtain new parameters from a single source network (Wang et al., 2023a; Xu et al., 2023). However, being fundamentally constrained to the spatial domain, these methods treat knowledge as a collection of discrete components (e.g., layers and neurons). This often makes their selection process fail to capture the holistic essence of the source model's knowledge.

These limitations inspired a novel paradigm called *learngene* (Wang et al., 2023b; Feng et al., 2025a), which offers an ideal theoretical goal. It posits that neural networks internalize a compact form of foundational knowledge – akin to genetic codes – that is not tied to specific architectures or tasks. If such a learngene could be successfully extracted and transferred, any downstream model could inherit

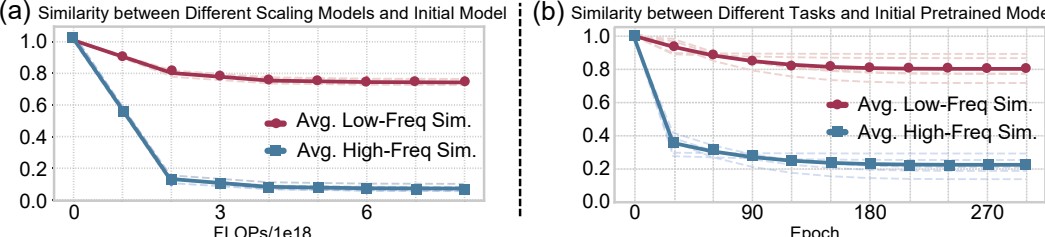

Figure 1: (a) Five different-scaled models initialized from the same DeiT are fine-tuned on the same downstream tasks. (b) A pre-trained DeiT is fine-tuned on five different downstream tasks. We plot the cosine similarity between the low-frequency components of the fine-tuned weights and their original pre-trained state over the training process. Each dashed line represents an individual fine-tuned model, while the solid line indicates their average.

a strong foundation and learn with extreme efficiency. However, existing learngene methods remain indirect and inefficient. Major approaches rely on selecting network fragments (Wang et al., 2022; 2023b), which risks preserving crucial parameter interdependencies and capturing only a fragmented representation of the true learngene. Some recent approaches involve training an auxiliary model from scratch, guided by strong prior assumptions (Xia et al., 2024; Feng et al., 2025b). The very need for a costly process contradicts the vision of a truly universal and efficient initialization. Consequently, there is a critical gap between concept and implementation.

Our core contribution is identifying the *low-frequency components of a network's weights* as a concrete carrier for the learngene concept. This insight is grounded in an empirical analysis of the spectral dynamics of parameter updates. We use the Discrete Cosine Transform (DCT), a signal processing tool for separating core information from high-frequency details (Wallace, 1991; Raid et al., 2014). We apply the DCT to decompose network weights into low- and high-frequency components and track their evolution in two scenarios: (a) fine-tuning of five different-scaled models that are initialized solely from the low-frequency components of that same DeiT model (Touvron et al., 2021) via inverse DCT (IDCT); (b) fine-tuning a single pre-trained DeiT model on five different downstream tasks. As illustrated in Figure 1, the low-frequency components consistently maintained remarkable stability and high similarity to their original source, while the high-frequency components proved highly volatile and task-specific. This finding suggests that low-frequency components could serve as the carrier of the achitecture- and task-agnostic knowledge that defines the learngene concept.

Building on this insight, we introduce FRONT (FRequency dOmain kNowledge Transfer), a novel framework that operationalizes the learngene concept by instantiating it as a low-frequency representation. FRONT employs the DCT to transform a pre-trained model's weights into the frequency domain, where the low-frequency coefficients are isolated as the learngene. A key advantage of FRONT is its flexibility. For plug-and-play deployment, the learngene can be directly *extracted from any off-the-shelf* pre-trained model, a process completed in *milliseconds on a CPU*. Alternatively, for seeking enhanced performance, we propose a *one-time refinement process*. This refinement can be applied either by training a model from scratch or, more efficiently, by briefly fine-tuning a pre-existing model within just a few epochs. It refines the source model with a novel spectral regularizer that penalizes high-frequency components, encouraging the model to discard task-specific details and preserve a more fundamental knowledge structure. Then, the learngene is adapted to a target *architecture of flexible depth or width* by simple truncation or padding before being transformed back into the spatial domain via the IDCT to yield the final initialized weights. This approach bridges a critical gap in the learngene's concept and operation by unlocking a previously unavailable capability: training-free, multi-size initialization from any pre-trained model.

Our extensive experiments validate the state-of-the-art performance and the efficiency of FRONT for initializing models across diverse settings. In vision tasks, models initialized with FRONT achieve the performance of a standard 150-epoch pre-training schedule in only 10 epochs, accelerating convergence by a factor of 15. This efficiency is mirrored in language tasks, where FRONT reduces the required training FLOPs by an average of 40.5% across all evaluated architectures compared to training from scratch. Furthermore, visualizations confirm that the extracted learngenes are highly structured, exhibiting patterns consistent with those observed in pre-trained models (Trockman & Kolter, 2023; Xu et al., 2023). By instantiating the learngene as low-frequency components in the DCT domain, we achieve a "one-for-all" parameter initialization that efficiently transfers knowledge from a single model to downstream models of various sizes.

## 2 RELATED WORK

**Model Initialization.** Effective weight initialization is a cornerstone of deep learning, evolving from early distribution-based methods like Xavier and He initialization (Glorot & Bengio, 2010; Chen et al., 2021) to the current paradigm of leveraging large-scale pre-trained models (He et al., 2016; Devlin et al., 2019). While pre-training provides a powerful starting point, it is challenging to efficiently adapt a single pre-trained model to target architectures of mismatched sizes. One line of work involves direct parameter manipulation; for instance, Wt Select (Xu et al., 2023) reuses weights by selecting parameter subsets from larger networks, while LiGO (Wang et al., 2023a) applies mathematical scaling for cross-size adaptation. Another direction employs complex generative models, such as GHN-3 (Knyazev et al., 2023), which uses a graph-based hypernetwork to synthesize entirely new parameters. Despite their ingenuity, these methods often suffer from high computational overhead, risk disrupting learned knowledge structures, or introduce parameter inconsistencies that can lead to negative transfer (Feng et al., 2025b). However, our approach preserves the core transferable knowledge from a source model and flexibly adapts it to various target sizes, enabling effective initialization without costly retraining.

**The DCT in Neural Networks.** The DCT is a fundamental technique in signal processing, renowned for its "energy compaction" property—its ability to concentrate the most significant information of a signal into a few low-frequency coefficients (Wallace, 1991). This property has made it the cornerstone of compression standards like JPEG (Raid et al., 2014). Recently, several studies have successfully applied the DCT to network weights, pruning high-frequency coefficients to reduce model memory usage with minimal impact on performance(Ulicny et al., 2022; 2021). However, these compression-focused methods implicitly assume that the preserved low-frequency knowledge is specific to a single task and architecture. We hypothesize and empirically demonstrate that these low-frequency coefficients represent the core, transferable knowledge—the *learngenes*—that can be seamlessly adapted to initialize a diverse array of target architectures across varying tasks.

**Learngene.** The learngene paradigm, inspired by biological genetics, proposes transferring compact, knowledge-rich network representation to initialize diverse architectures (Feng et al., 2025a; Wang et al., 2023b; Feng et al., 2025b). Some methods, such as HeurLG (Wang et al., 2022) and AutoLG (Wang et al., 2023b), typically rely on either heuristic-based selection of fragments, using criteria like gradient stability or representation similarity, or on multi-stage linear expansion, where TLEG (Xia et al., 2024) and WAVE (Feng et al., 2025b) require specialized auxiliary training guided by strong prior assumptions. The reliance on such discrete, structurally rigid fragments or expensive retraining contradicts the Learngene vision of a truly universal and efficient initialization paradigm. In contrast, FRONT directly utilizes full pre-trained parameters, which enables the initialization of models with varying sizes in a single and unified operation.

## 3 METHODS

### 3.1 PRELIMINARY

DCT efficiently converts signals to frequency-domain representations. While 1D/2D-DCT are prevalent in traditional signal/image processing, the 3D-DCT/IDCT variant is adopted here to handle the tensor-structured weight matrices in neural networks. The transformation formulas for other dimensions and more detailed formulas are shown in the Appendix A.2.

**3D-DCT** For the weight $x \in \mathbb{R}^{M \times N \times P}$, the transformed coefficient matrix $X \in \mathbb{R}^{M \times N \times P}$ is:

$$X[k,l,q] = \mathcal{D}(x) = \alpha(k)\alpha(l)\alpha(q) \sum_{m=0}^{M-1} \sum_{n=0}^{N-1} \sum_{p=0}^{P-1} x[m,n,p]\, C(m,n,p,k,l,q). \tag{1}$$

**3D-IDCT** The IDCT recovers the original signal:

$$x[m,n,p] = \mathcal{D}^{-1}(X) = \alpha(k)\alpha(l)\alpha(q) \sum_{k=0}^{M-1} \sum_{l=0}^{N-1} \sum_{q=0}^{P-1} X[k,l,q]\, C(m,n,p,k,l,q). \tag{2}$$

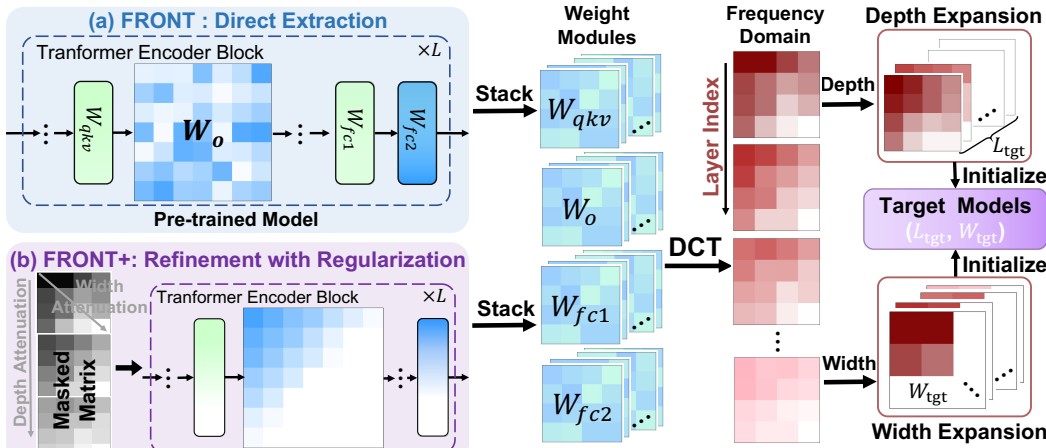

Figure 2: **The overview of FRONT and FRONT+.** The learngenes are obtained from either FRONT via the open available pre-trained models or FRONT+ using source models refined with our frequency regularization. The core process involves transforming weight modules via DCT to extract task-agnostic knowledge. Finally, these learngenes are used to initialize diverse target models through the IDCT, which flexibly accommodate varying architectures via zero-padding or truncation.

## 3.2 OVERVIEW OF OUR METHOD

Our method extends frequency-domain knowledge to the weight spaces—exemplified here with Vision Transformers (ViTs)—through 3D-DCT weight transformations, as shown in Figure 2. Task-agnostic knowledge within the model's parameters is predominantly encoded in the low-frequency components, hereafter referred to as *learngenes*. Notably, the applicability of this approach extends beyond transformer-based models to include other architectures (see details in Appendix B), such as Multi-Layer Perceptrons (MLPs) and Convolutional Neural Networks (CNNs).

The learngenes can be extracted using two distinct strategies, detailed in Sections 3.3 and 3.4, offering a trade-off between efficiency and performance. **(1) FRONT**: Direct extraction immediately harvests low-frequency coefficients from any off-the-shelf pre-trained model for instantaneous, zero-cost initialization. **(2) FRONT+**: Refinement employs a frequency regularizer to generate more effective learngenes, a process that can be implemented from scratch or, more efficiently, through a brief fine-tuning of a pre-existing model, often *in just a few epochs*. Regardless of the chosen strategy, the resulting learngenes serve as a universal blueprint, enabling rapid and flexible initialization of diverse target models in milliseconds on a CPU via IDCT-based reconstruction in Section 3.5.

## 3.3 FRONT: DIRECT EXTRACT LEARNGENES FROM PRE-TRAINED MODELS

The direct extraction strategy operates directly on open available pre-trained models, eliminating the computational overhead of any additional training. For ViTs, weight matrices are collected from all $L$ transformer layers–including the queries, keys, and values ($W_{qkv}^{(1\sim L)}$), attention output ($W_o^{(1\sim L)}$), and two feed-forward layers ($W_{fc1}^{(1\sim L)}, W_{fc2}^{(1\sim L)}$)–as

$$\Omega = \{W_{qkv}, W_o, W_{fc1}, W_{fc2}\} \subset \mathbb{R}^{L \times d_{\text{in}} \times d_{\text{out}}}, \tag{3}$$

where $L$ is the total number of transformer layers, and $d_{\text{in}}, d_{\text{out}}$ are input and output dimensions.

Based on the finding that a model's foundational knowledge is concentrated in its low-frequency coefficients shown in Figure 1, we extract these **learngenes** as follows:

**DCT Transformation:** Each $W^i \in \Omega$ is transformed into its frequency domain using 3D-DCT: $\Phi^i = \mathcal{D}(W^i)$, where the resulting $\Phi^i$ preserves the dimensions of the original weight.

**Preservation and Truncation:** A binary mask $\mathbf{M}_r$ is applied to preserve only the low-frequency coefficients along all three dimensions (layer, input, output), defined by the frequency ratio $r$:

$$\mathcal{G}^i = M_r \odot \Phi^i, \quad M_r[l,m,n] = \begin{cases} 1 & \text{if } l \le \lfloor rL \rfloor, \ m \le \lfloor rd_{\text{in}} \rfloor, \ n \le \lfloor rd_{\text{out}} \rfloor \\ 0 & \text{otherwise} \end{cases}, \tag{4}$$

where $\odot$ denotes truncating the coefficient values of zero after dot multiplication. This step yields a compact set of coefficients, which we define as the **learngenes** $\mathcal{G}$. By discarding high-frequency information, this approach significantly reduces computing requirements compared to high-effort transfer while preserving essential knowledge. This makes direct extraction a scalable and lightweight solution for initializing downstream models.

### 3.4 FRONT+: EXTRACTION VIA REFINEMENT WITH FREQUENCY REGULARIZATION

While direct extraction is efficient, its reliance on hard truncation presents two key limitations. First, the abrupt cutoff can introduce spectral artifacts, disrupting the smoothness of the learned representations. Second, by completely discarding high-frequency coefficients, this approach risks eliminating fine-grained details that may contribute to the overall coherence of the knowledge.

To overcome these limitations, we introduce a refinement strategy that employs a smooth frequency regularization, which can be implemented from scratch or, fine-tuning a pre-trained model within minimal computational overhead. Before the hard cutoff, this method progressively suppresses high-frequency components while allowing knowledge to converge towards lower frequencies. Specifically, we penalize the energy of high-frequency coefficients during the model's optimization process:

$$\mathcal{L}_{\text{reg}} = \sum_{\Phi^i \in \Omega} \left( \frac{1}{\|M_{r+} \odot \Phi^i\|_0} \sum_{l,m,n} (M_{r+}[l,m,n] \cdot \Phi^i[l,m,n])^2 \right), \tag{5}$$

where, instead of a binary mask, $\mathbf{M}_{r+}$ is a soft, dimension-wise penalty mask that assigns higher weight to higher frequencies:

$$M_{r+}[l,m,n] = \prod_{d \in \{l,m,n\}} \left[ 1 - \exp\left( -\frac{f_d}{\gamma_d} \right) \right], \tag{6}$$

where $f_l = \frac{l}{L}$, $f_m = \frac{m}{d_{\text{in}}}$, $f_n = \frac{n}{d_{\text{out}}}$ represent normalized frequency indices, and $\gamma_d$ are decay-rate hyperparameters. This formulation maintains gradient updates through all frequencies during refinement training while progressively attenuating high-frequency components. The total loss combines a task-specific objective $\mathcal{L}_{\text{total}}$ with our frequency regularization:

$$\mathcal{L}_{\text{total}} = (1 - \lambda)\mathcal{L}_{\text{task}} + \lambda \mathcal{L}_{\text{reg}}, \tag{7}$$

where $\mathcal{L}_{\text{task}}$ is the primary task training objective, such as cross-entropy for image classification or language modeling loss for pre-training transformers. Certainly, this term can be incorporated with other capability techniques, such as distillation loss (Hinton et al., 2015), as detailed in Appendix B.1. After the refinement process, we extract the learngenes $\mathcal{G}$ using the same DCT and preservation steps outlined in Eq.( 4 ). By progressively attenuating high frequencies, this approach generates smoother and more robust learngenes, which serve as a more effective foundation for initializing downstream models with varying sizes.

### 3.5 INITIALIZATION OF VARIABLE-SIZED MODELS

A key advantage of our framework is its ability to initialize target models of arbitrary sizes using a single set of learngenes. The extracted learngenes $\mathcal{G}$ can be obtained through direct extraction by FRONT, or through refinement with frequency regularization by FRONT+. For target models with layers $L_{\text{tgt}}$ and hidden dimensions $d_{\text{tgt}}$, we adapt the learngenes through a resizing operation entirely in the frequency domain, which can be performed *in mere milliseconds on a CPU*:

$$\Phi_{\text{tgt}}^i = \mathcal{P}_{\text{pad}}\left(\mathcal{G}^i\right) \text{ or } \mathcal{P}_{\text{trunc}}\left(\mathcal{G}^i\right) \in \mathbb{R}^{L_{\text{tgt}} \times d_{\text{tgt}}^{\text{in}} \times d_{\text{tgt}}^{\text{out}}}, \tag{8}$$

where $\mathcal{P}_{\text{pad}}$ applies zero padding to the high-frequency regions of the spectrum to increase tensor dimensions, while $\mathcal{P}_{\text{trunc}}$ discards the outermost high-frequency coefficients to reduce them. The final weights are then reconstructed via the 3D-IDCT as $W_{\text{tgt}}^i = \mathcal{D}^{-1}(\Phi_{\text{tgt}}^i) \in \mathbb{R}^{L_{\text{tgt}} \times d_{\text{tgt}}^{\text{in}} \times d_{\text{tgt}}^{\text{out}}}$.

We provide an efficient, flexible framework for knowledge transfer by extracting knowledge into learngenes—via zero-overhead direct extraction or frequency-regularized refinement. Once learngenes are extracted, they are fully decoupled from the original models and enables initialization for models of varying depths and widths. Our approach overcomes conventional transfer limitations, promoting task-agnostic knowledge adaptation across diverse architectures.

# 4 EXPERIMENTS

## 4.1 EXPERIMENTAL SETUP

In this section, we systematically evaluate the effectiveness of our method across different architectures and tasks, organizing our analysis by modality. For vision models, we first assess scalability by varying the depth and width of DeiT in Section 4.2.1, including a thorough 300-epoch analysis of direct initialization quality. We then examine cross-dataset and cross-task generalization in Section 4.2.2. For language models, we conduct analogous scaling experiments and evaluate downstream performance on the GLUE benchmark for models like BERT in Section 4.3. Finally, Section 4.4 presents ablation and analyses to understand the underlying mechanisms that make FRONT+ effective.

**Datasets.** For vision tasks, all source models, whether used for direct extraction or refinement, are pre-trained on ImageNet-1K. This dataset is also used for our vision scaling experiments. We assess cross-dataset transferability on a diverse suite of seven downstream classification datasets. Furthermore, we evaluate generalization to other vision domains, using six datasets for object detection and four for image segmentation. For language tasks, we conduct training on standard large-scale corpora: the English Wikipedia corpus is used for BERT (Devlin et al., 2019) and RoBERTa (Liu et al., 2019), while the concatenation of English Wikipedia and the Toronto Book Corpus is used for GPT-2 (Liu et al., 2019). The downstream performance of our initialized language models is subsequently evaluated on the GLUE benchmark (Wang et al., 2018).

**Architectural Details.** Our evaluation spans a wide range of architectures in both vision and language domains. For vision models, we utilize publicly available 12-layers DeiT-Ti/S/B models pre-trained on ImageNet-1K as the source for direct extraction. For our refinement, we train three compact 8-layer auxiliary models (with 3, 6, or 12 heads) from scratch for 150 epochs on ImageNet-1K. We also explore a more efficient refinement variant by briefly fine-tuning the pre-trained DeiT models. To assess scalability, we initialize DeiT variants by varying their depth (4–12 layers) and width (6–24 heads; 384–1536 dimensions). For generalization experiments, we initialize ResNet-50/152 from a ResNet-101 source, and also use ViT-S/B and ResNet-50 as backbones for downstream object detection and semantic segmentation tasks, respectively. For language models, we demonstrate cross-scale transfer by initializing a 6-layer, 384-dimension model (e.g., BERT-S) using learngenes extracted from its corresponding 12-layer, 768-dimension Base counterpart (BERT-B). This procedure is applied across three foundational architectures: BERT, RoBERTa, and GPT-2. For more experimental details, including visual and language models and datasets, please refer to Appendix C.

**Baseline.** To provide a comprehensive evaluation for vision tasks, we benchmark our framework's flexible learngene sourcing strategies against two main categories of initialization methods: **(1) Direct Initialization.** Methods here rely on prior knowledge or direct parameter transfer applied to existing pre-trained models on ImageNet-1K without additional training. This includes He-Init (Chen et al., 2021), Mimetic-Init (Trockman & Kolter, 2023), Wt Select (Xu et al., 2023), Heur-LG (Wang et al., 2022), Cluster-LG (Wang et al., 2024b), LiGO (Wang et al., 2023a), and **FRONT**. **(2) Methods with Extra Training.** This category encompasses methods that require a dedicated optimization phase to generate transferable knowledge. Methods like GHN-3 (Knyazev et al., 2023), Share-Init (Lan et al., 2019), Auto-LG (Wang et al., 2023b), TELG (Xia et al., 2024), and WAVE (Feng et al., 2025b), execute a dedicated, computationally intensive process to generate or discover transferable parameters, a necessity as their frameworks typically do not support the direct fine-tuning of the pre-trained models. To ensure the most direct and rigorous comparison against these high-effort methods, our main experiments also adopt the from-scratch paradigm with **FRONT+**. We use the same pretraining data and experimental settings for all the methods for a fair comparison. Notably, our refinement framework also supports a highly efficient fine-tuning strategy (**FRONT++**). Our preliminary results indicate that this approach can surpass the from-scratch version in performance with significantly fewer training cost, detailed in Section 4.4.1. For language tasks, we established two baselines: training the Small model from scratch and employing knowledge distillation (Hinton et al., 2015), where the respective Base model served as the teacher.

**Evaluation Metrics.** The main metric is Top-1 accuracy, measuring initialization effectiveness. Supplementary metrics are convergence efficiency (epochs) and parameter transfer efficiency.

Table 1: Performance of models initialized with different depths on ImageNet-1K for 10 epochs. Para.(M) denotes the number of parameters, specified per model size (rows) and as the average transferred during initialization (columns). "Epoch" indicates the **extra training epochs** for knowledge merging or pre-training. " / " denotes failed initialization, and "N/A" indicates that the metric is not applicable. The **best** result in each column is highlighted in bold, while the second-best is underlined.

| | Methods | Epoch | Para. | $W_{192}$ (DeiT-Ti) $L_4$ | $L_6$ | $L_8$ | $L_{10}$ | $L_{12}$ | Para. | $W_{384}$ (DeiT-S) $L_4$ | $L_6$ | $L_8$ | $L_{10}$ | $L_{12}$ | Para. | $W_{768}$ (DeiT-B) $L_4$ | $L_6$ | $L_8$ | $L_{10}$ | $L_{12}$ |
|---|---|---|---|---|---|---|---|---|---|---|---|---|---|---|---|---|---|---|---|---|
| | | | | 2.2 | 3.1 | 4.0 | 4.9 | 5.8 | | 7.9 | 11.5 | 15.0 | 18.6 | 22.2 | | 29.9 | 44.1 | 58.3 | 72.5 | 86.7 |
| Direct | He-Init | 0 | 0 | 34.7 | 40.6 | 43.7 | 46.8 | 48.3 | 0 | 42.2 | 49.4 | 52.1 | 53.7 | 55.5 | 0 | 47.9 | 53.1 | 54.4 | 55.0 | 56.7 |
| | Mimetic | 0 | 0 | 35.1 | 40.2 | 43.2 | 46.3 | 48.1 | 0 | 43.3 | 49.1 | 53.0 | 54.1 | 55.6 | 0 | 50.2 | 54.3 | 56.5 | 58.5 | 58.6 |
| | Wt Select | 0 | 3.9 | 46.1 | 52.3 | 55 | 56.8 | 58.9 | 15.0 | 50.1 | 56 | 57.5 | 61.9 | 63.3 | 58.2 | 56.7 | 61.1 | 63.6 | 64.8 | 65.9 |
| | Heur-LG | 0 | 1.7 | 41.5 | 47.4 | 50.5 | 53.5 | 55.5 | 6.1 | 52.3 | 57.3 | 61.7 | 64.4 | 65.9 | 22.8 | 60.5 | 68.7 | 72.2 | 73.6 | 74.0 |
| | Cluster-LG | 0 | 2.4 | 46.6 | 51.5 | 51.9 | 55.8 | 57.2 | 5.1 | 53.2 | 54.9 | 52.6 | 52.1 | 55.5 | 19.6 | 52.7 | 60.4 | 60.6 | 58.6 | 62.7 |
| | LiGO | 0 | 2.2 | / | 59.0 | 60.2 | 59.8 | 60.9 | 7.9 | / | 68.6 | 69.9 | 69.7 | 70.0 | 29.9 | / | 74.2 | 74.4 | 75.3 | 75.4 |
| | FRONT | 0 | 2.2 | 55.3 | 63.3 | 64.4 | 64.7 | 65.3 | 8.1 | 63.4 | 68.9 | 70.5 | 70.9 | 71.2 | 32.4 | 72.1 | 74.6 | 75.3 | 75.8 | 76.3 |
| Train | GHN-3 | N/A | N/A | 40.9 | 45.0 | 46.6 | 49.1 | 48.9 | N/A | 45.4 | 49.0 | 50.2 | 52.3 | 53.2 | N/A | 49.5 | 52.5 | 53.8 | 54.2 | 54.3 |
| | Share Init | 150 | 0.8 | 55.2 | 59.8 | 62.5 | 64.3 | 65.3 | 2.5 | 65.0 | 69.7 | 71.7 | 72.7 | 73.3 | 8.6 | 71.7 | 75.3 | 76.4 | 77.4 | 77.6 |
| | Auto-LG | 50 | 2.2 | 52.4 | 61.8 | 64.6 | 65.9 | 66.8 | 7.9 | 63.2 | 70.5 | 72.2 | 73.3 | 73.8 | | 60.9 | 70.0 | 72.4 | 73.5 | 73.8 |
| | TLEG | 150 | 1.3 | 55.0 | 60.5 | 62.9 | 64.4 | 65.4 | 4.3 | 65.4 | 70.5 | 72.1 | 73.2 | 73.8 | 15.7 | 71.6 | 74.9 | 76.2 | 77.0 | 77.1 |
| | WAVE | 150 | 1.3 | 58.6 | 63.2 | 65.4 | 66.6 | 67.3 | 4.3 | 68.9 | 72.7 | 74.1 | 74.9 | 75.3 | 15.7 | 74.5 | 77.5 | 78.2 | 78.9 | 79.2 |
| | FRONT+ | 150 | 0.8 | 58.7 | 63.4 | 65.6 | 66.8 | 67.5 | 3.2 | 68.9 | 72.9 | 74.2 | 75.0 | 75.4 | 13.0 | 74.5 | 77.7 | 78.5 | 79.0 | 79.3 |
| PT | Direct PT | 150×15 | 4.0 | 50.4 | 57.7 | 62.7 | 66.2 | 68.6 | 15.0 | 62.6 | 70.1 | 73.8 | 76.0 | 77.6 | 58.3 | 70.7 | 76.2 | 79.1 | 80.5 | 81.5 |

## 4.2 RESULTS ON VISION TASKS

### 4.2.1 INITIALIZATION ABILITY ON VARIABLE-SIZED MODELS

**Depth Expansion.** We systematically evaluate the initialization of variable-sized models in Table 1, a crucial requirement for resource-constrained deployment, by benchmarking performance on ImageNet-1K. Our method consistently outperforms both direct processing and learngene approaches. Methods leveraging pre-trained models surpass training from scratch, highlighting the benefits of leveraging prior knowledge. While LiGO improves over Mimetic, its injection of random parameters brings to model size mismatch. In contrast, FRONT exploits low-frequency knowledge from pre-trained models, surpassing LiGO by 2% and He-Init by 19.8%.

Table 2: Performance of initializing models with variable widths on ImageNet-1K.

| | Methods | Epoch | Para. | $L_6$ $W_{384}$ | $W_{576}$ | $W_{768}$ | $W_{1152}$ | $W_{1536}$ |
|---|---|---|---|---|---|---|---|---|
| | | | | 11.5 | 25.1 | 44.1 | 98.0 | 173.1 |
| Direct | He-Init | 0 | 0 | 49.4 | 51.5 | 53.1 | 46.6 | 31.3 |
| | Mimetic | 0 | 0 | 49.1 | 48.0 | 54.3 | 47.7 | 33.0 |
| | Wt Select | 0 | 26.8 | 48.6 | 50.7 | 55.4 | / | / |
| | LiGO | 0 | 10.0 | 63.6 | 63.1 | 69.5 | 70.0 | 73.7 |
| | FRONT | 0 | 8.9 | 62.3 | 63.6 | 69.8 | 70.7 | 73.9 |
| Train | GHN-3 | N/A | N/A | 49.0 | 51.8 | 52.5 | 52.7 | 51.0 |
| | WAVE | 150 | 5.4 | 64.8 | 67.0 | 72.4 | 73.5 | 78.3 |
| | FRONT+ | 150 | 5.5 | 66.3 | 68.3 | 73.5 | 74.6 | 78.7 |

Compared to methods requiring additional training, Share Init outperforms GHN3 by combining rule-based priors with pre-trained layers. In contrast, several learngene methods like Auto-LG and TLEG impose rigid layer-specific constraints, limiting scalability across model configurations. While WAVE performs competitively via weight templates, FRONT+ surpasses it while transferring over 25% fewer parameters. Notably, FRONT+ outperforms 150-epoch pretraining after only 10 epochs, highlighting its efficiency and flexibility. We also demonstrated our training curve in Figure 9.

**Width Expansion.** The reversible nature of DCT and IDCT enables FRONT to flexibly scale model widths. As shown in Table 2, our method consistently outperforms other initialization and transfer methods. Direct initialization methods like Mimetic exhibit substantially lower performance, underscoring the critical role of pre-trained knowledge in effective model initialization. LiGO introduces excessive randomness, hindering the initialization of larger models, while Wt Select disrupts structural knowledge transfer. Although WAVE utilizes Kronecker operations with

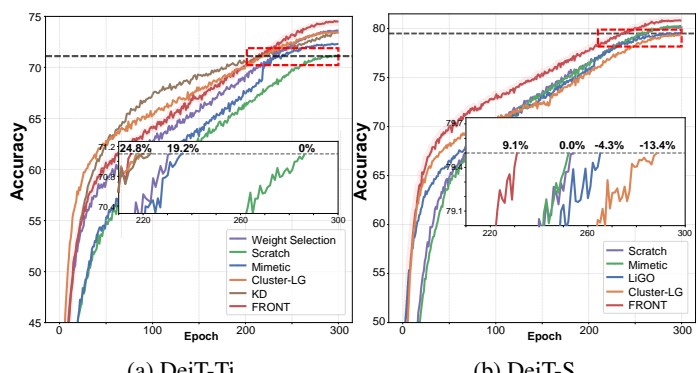

(a) DeiT-Ti          (b) DeiT-S

Figure 3: Comparison of different direct initialization methods during full training on DeiT-Ti and DeiT-S.

Table 3: Performance of models on downstream datasets. "Para.(M)" is the average parameter transferred during model initialization.

| | Methods | DeiT-Ti, 3.0M | | | | | | | | | DeiT-S, 11.3M | | | | | | | | |
|---|---|---|---|---|---|---|---|---|---|---|---|---|---|---|---|---|---|---|---|
| | | Para. | Flow. | CUB | Cars | C10 | C100 | Food | iNat. | Aver. | Para. | Flow. | CUB | Cars | C10 | C100 | Food | iNat. | Aver. |
| Direct | He-Init | 0 | 53.9 | 26.1 | 19.9 | 92.4 | 68.3 | 68.4 | 52.3 | 54.5 | 0 | 57.2 | 27.3 | 23.8 | 94.0 | 66.5 | 70.6 | 54.0 | 56.2 |
| | Mimetic | 0 | 52.1 | 35.0 | 20.5 | 88.9 | 63.4 | 66.9 | 49.0 | 53.7 | 0 | 57.4 | 39.6 | 34.2 | 91.6 | 65.7 | 67.1 | 52.2 | 58.3 |
| | Wt Select | 2.9 | 55.0 | 34.4 | 21.7 | 92.5 | 67.0 | 67.4 | 51.2 | 55.6 | 11.0 | 58.3 | 33.1 | 28.1 | 94.1 | 68.1 | 69.0 | 54.2 | 57.8 |
| | Heur-LG | 1.5 | 64.7 | 44.6 | 37.7 | 94.0 | 71.1 | 74.7 | 57.4 | 63.5 | 5.7 | 69.1 | 48.0 | 51.2 | 95.1 | 72.8 | 76.8 | 59.3 | 67.5 |
| | Cluster-LG | 2.4 | 62.8 | 44.8 | 36.9 | 94.1 | 74.5 | 78.6 | 56.4 | 64.0 | 5.1 | 65.5 | 45.4 | 46.3 | 95.4 | 74.6 | 78.4 | 60.4 | 66.6 |
| | LiGO | 2.0 | **94.2** | **71.8** | **83.9** | 95.6 | 78.5 | 82.1 | 61.6 | *81.1* | 7.5 | **95.9** | **74.8** | **87.9** | 96.9 | 81.3 | 84.0 | 66.1 | *83.8* |
| | FRONT | 2.2 | 92.9 | 70.5 | 82.3 | **95.9** | **78.7** | **83.7** | **63.7** | *81.1* | 8.1 | 94.5 | 74.1 | 87.4 | **97.2** | **81.8** | **85.2** | **66.6** | *83.9* |
| Train | GHN-3 | N/A | 50.0 | 41.1 | 23.2 | 92.5 | 70.1 | 76.2 | 51.7 | *57.8* | N/A | 52.7 | 45.2 | 30.6 | 93.9 | 72.7 | 76.2 | 55.5 | *61.0* |
| | Share Init | 0.6 | 92.4 | 70.1 | 82.1 | 96.0 | 77.2 | 81.2 | 63.2 | *80.3* | 2.2 | 94.1 | 72.4 | 87.2 | 96.5 | 78.5 | 83.0 | 63.0 | *82.1* |
| | Auto-LG | 2.0 | 93.5 | 71.4 | 83.5 | 96.4 | 77.1 | 81.7 | 62.5 | *80.9* | 7.5 | 96.4 | 75.1 | 88.2 | 97.3 | 81.0 | 84.6 | 67.0 | *84.2* |
| | TLEG | 1.1 | 91.0 | 69.5 | 78.2 | 96.1 | 77.0 | 82.0 | 63.4 | *79.6* | 3.9 | 93.7 | 72.6 | 87.2 | 97.2 | 80.2 | 84.9 | 66.5 | *83.2* |
| | WAVE | 1.1 | 94.9 | **74.8** | 84.4 | 96.6 | 80.7 | 83.8 | 65.2 | *82.9* | 4.0 | 96.9 | 78.1 | 89.4 | 97.4 | 83.2 | 85.5 | 67.6 | *85.4* |
| | FRONT+ | 0.8 | **95.1** | **75.2** | **86.1** | **96.6** | **80.8** | **83.9** | **65.4** | *83.3* | 3.2 | **97.2** | **78.2** | **89.4** | **97.4** | **84.0** | **86.1** | **68.1** | *85.8* |
| FT | Direct FT | 2.9 | 95.4 | 75.1 | 86.5 | 96.6 | 80.2 | 84.0 | 66.9 | *83.5* | 11.0 | 96.4 | 77.0 | 89.4 | 97.5 | 82.8 | 85.6 | 69.3 | *85.4* |

Table 4: Object Detection (ViT-10shot) SD: COCO → TD: Below

| Methods | ViT-S | | | | | | | ViT-B | | | | | | |
|---|---|---|---|---|---|---|---|---|---|---|---|---|---|---|
| | ArTaxOr | Clipart1k | DIOR | DeepFish | NEU-DET | UODD | Aver. | ArTaxOr | Clipart1k | DIOR | DeepFish | NEU-DET | UODD | Aver. |
| Scratch | 41.97 | 27.70 | 21.40 | 20.00 | 12.57 | **5.15** | *21.47* | 51.42 | 37.59 | 24.50 | 19.22 | 12.99 | **5.81** | *25.26* |
| FRONT | **42.36** | **29.74** | **23.33** | **20.73** | **12.71** | 5.09 | *22.33* | **53.48** | **39.49** | **25.48** | **19.71** | **13.21** | 5.78 | *26.19* |

weight templates for reasonable results, FRONT+'s transformation of low-frequency information further enhances source models' decoupling. This allows our method to achieve robust and flexible initialization across varying model widths while preserving essential knowledge structures.

**Initialization Performance during Full Training.** Figure 3 shows the training performance of various direct initialization methods between DeiT-S and DeiT-Ti over 300 epochs. All reported results for FRONT are averaged over five independent runs with different random seeds to ensure robustness. Methods leveraging pre-trained knowledge, including Wt Select, LiGO, Knowledge Distillation (KD) and Cluster-LG, exhibit stronger early-stage performance than random initialization. In contrast, FRONT employs a unified conversion strategy that preserves low-frequency components, enabling effective initialization and consistently strong performance throughout training.

### 4.2.2 GENERALIZATION ACROSS DOWNSTREAM DATASETS.

**Initialization on DeiT-based Architectures.** In Table 3, all methods, including ours, use a source model pre-trained on ImageNet-1K for initialization. Following that, the downstream models are directly trained on downstream tasks without any additional pre-training. We supplement the additional experiment of FRONT on the ResNet models in Appendix D.2.

Our method consistently improves performance across all datasets, demonstrating strong generalization from initialization. In contrast, methods such as Mimetic and GHN-3 show limited adaptability on certain datasets, while Wt Select occasionally underperforms He-Init on specific datasets like iNaturalist-2019. These results emphasize the need to transfer essential knowledge without task overfitting. Notably, FRONT+ achieves superior performance while transferring fewer parameters than WAVE. By efficiently leveraging low-frequency components, our method reduces storage overhead and enables flexible, effective initialization.

**Initialization on Cross-Domain Tasks.** To further verify the capability of FRONT in extracting generalizable knowledge, we follow the experimental protocols introduced in (Fu et al., 2024) for object detection and (Nie et al., 2024) for image segmentation from Source Domain (SD) to Target Domain (TD). Specifically, we evaluate our approach on two architectures, including ViT-S/B and ResNet50, using six object detection datasets and four image segmentation datasets, comparing the FRONT initialization with the scratch training, other details in Appendix C.2.4.

Table 5: Image Segmentation (ResNet50-5shot) SD: ImageNet-1K→TD: Below

| Dataset | Scratch | FRONT |
|---|---|---|
| Deepglobe | 28.32 | **44.31** (↑ 15.99) |
| ISIC | 58.97 | **63.90** (↑ 4.93) |
| Chest X-Ray | 38.56 | **67.29** (↑ 28.73) |
| FSS-1000 | 43.06 | **66.44** (↑ 23.38) |
| Avg. | 42.23 | **60.49** (↑ 18.26) |

As shown in Table 4 and Table 5, the proposed FRONT initialization substantially improves performance across the evaluated cross-domain tasks compared to training from scratch. Especially in image segmentation, FRONT achieves an average performance improvement of 18.26%. These

results demonstrate the effectiveness of extracting and transferring task-agnostic knowledge, leading to consistently higher accuracy in both object detection and image segmentation tasks.

## 4.3 RESULTS ON LANGUAGE TASKS

**More Effective Pre-training.** Across all three architectures, models initialized with FRONT demonstrate a markedly faster convergence presented in Figure 4. This effect is particularly dramatic for BERT, where FRONT also provides a significantly lower starting loss, and this performance lead is maintained throughout the training. While KD also leverages pre-trained knowledge, it incurs a substantial, cumulative computational cost from repeated teacher inferences at every training step. In contrast, FRONT imparts its full knowledge benefit in a single, zero-overhead initialization, leading to significant computational savings. On average, our method reduces the required training FLOPs by 40.5% across the three architectures compared to standard from-scratch training, underscoring its superiority as a resource-efficient knowledge transfer mechanism.

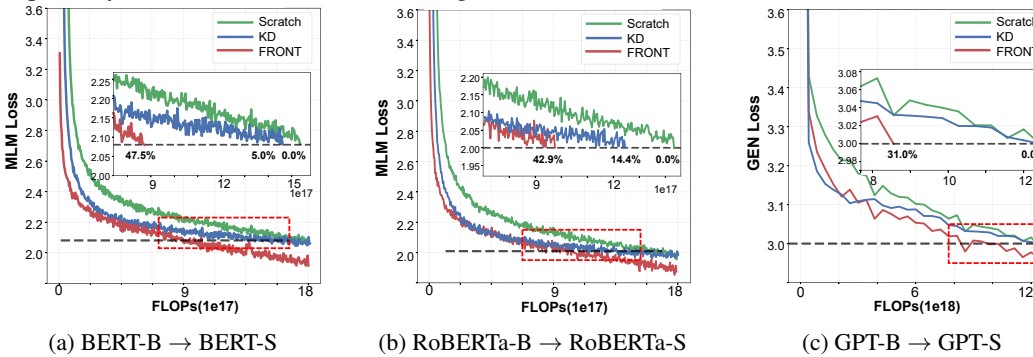

| (a) BERT-B → BERT-S | (b) RoBERTa-B → RoBERTa-S | (c) GPT-B → GPT-S |
|---|---|---|

Figure 4: Results of pretraining BRET-S, RoBERTa-S and GPT-S. FRONT can achieve the highest savings in FLOPs with 47.5% for BERT-S, 42.9% for RoBERTa-S, and 31.0% for GPT-S from the Scratch models.

**Generalization on GLUE Benchmark.** As presented in Table 6, the model initialized by FRONT significantly outperforms both the scratch and knowledge distillation baselines across all GLUE tasks. These results demonstrate that our method possesses strong generalization capabilities. It effectively transfers the core competencies acquired during unsupervised pre-training—such as the fundamental ability to predict masked tokens—which readily adapt to a diverse set of specific downstream tasks. Overall, these findings confirm that FRONT provides a robust and consistent initialization method that is compatible with multiple model architectures.

Table 6: GLUE benchmark results on BERT-S.

| Task | Scratch | KD | FRONT |
|---|---|---|---|
| SST-2 | 78.97 | 77.75 | **82.44** (↑ 3.47) |
| MNLI | 63.65 | 64.81 | **74.08** (↑ 9.27) |
| MRPC | 66.42 | 64.14 | **68.46** (↑ 2.04) |
| CoLA | 8.14 | 9.14 | **17.38** (↑ 8.24) |
| QNLI | 57.21 | 58.00 | **70.86** (↑ 13.65) |
| QQP | 81.50 | 80.07 | **84.93** (↑ 4.86) |
| STS-B | 17.27 | 15.76 | **54.23** (↑ 36.96) |
| Average | 53.31 | 52.81 | **64.63** (↑ 11.32) |

## 4.4 ABLATION AND ANALYSIS

### 4.4.1 BALANCE BETWEEN REFINEMENT DURATION AND REGULARIZATION STRENGTH

To show the ability of FRONT++ strategy, we conduct a grid search by fine-tuning a DeiT-Ti-L8 source model on ImageNet-1K, exploring the interplay between fine-tuning duration across various epochs $\{10, 20, 50, 100\}$ and regularization strength $\lambda$ values $\{1e-5, 5e-4, 1e-3, 2e-3, 5e-3\}$. Subsequently, the effectiveness of the 0.8M-parameter learngenes generated from each configuration is evaluated to initialize a deeper DeiT-Ti-L10 target model.

The results, presented in Figure 5, reveal the remarkable efficiency of this approach. A mere 20 epochs of fine-tuning yields an initialization capability comparable to the 150-epoch from-scratch FRONT+ variant, drastically reducing the required computation while still outperforming high-effort methods like WAVE. A clear trade-off

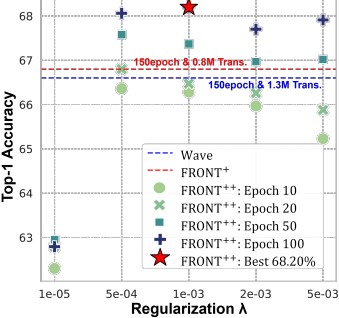

Figure 5: Achieving Superior Performance with Efficient Refinement.

also exists for $\lambda$: both insufficient (e.g., 1e-5) and excessive (e.g., 5e-3) regularization degrade performance, indicating a balance between concentrating core knowledge and preserving useful high-frequency details. These findings establish FRONT++ as a highly practical, resource-efficient solution for adapting pre-trained knowledge, achieving superior efficacy as a low-effort alternative.

### 4.4.2 EFFECT OF FREQUENCY-RATIO AND DECAY FACTOR

To investigate the influence of the low-frequency retention ratio $r$ and the refinement decay factor $\gamma_d$, we perform a ablation study using ImageNet-1K classification on DeiT-Ti_L8 as the source model. The frequency ratio $r$ denotes the proportion of DCT coefficients retained as the *learngene* from the source network parameters. The decay factor $\gamma_d$, as defined in Eq. ( 6 ), controls the suppression strength of high-frequency components during the refinement stage.

Across all $r$ values, the proposed FRONT consistently outperforms the *Scratch* baseline by a large margin ($\geq$ 13.0+%), confirming the robustness of low-frequency parameter transfer. Performance gains plateau when $r \in [0.50, 0.83]$ and $\gamma_d \in [0.125, 0.25]$, indicating that moderate refinement effectively attenuates harmful high-frequency noise without excessively discarding potentially useful fine-grained details. Extreme values of $r$ or $\gamma_d$ yield little additional benefit, suggesting that hyperparameter tuning can be coarse-grained for practical deployment in resource-limited environments.

Table 7: Top-1 accuracy (%) on ImageNet-1K as a function of $r$ and $\gamma_d$. And '−' corresponds to a direct application of FRONT without refinement.

| $\gamma_d \setminus r$ | 0.33 | 0.50 | 0.67 | 0.83 | 1.00 |
|---|---|---|---|---|---|
| – | 60.63 | 62.74 | 64.79 | 65.06 | 66.19 |
| $^1/_{16}$ | 66.61 | 66.64 | 66.65 | 66.58 | 66.60 |
| $^1/_8$ | 66.97 | 67.22 | 67.02 | 67.11 | 67.13 |
| $^1/_4$ | 67.08 | 67.58 | 68.02 | 67.92 | 67.88 |
| $^1/_2$ | 66.74 | 66.92 | 66.00 | 66.72 | 66.94 |
| 1 | 66.57 | 66.76 | 66.60 | 66.69 | 66.75 |

### 4.4.3 VISUALIZATION OF STRUCTURE AND COMMON KNOWLEDGE

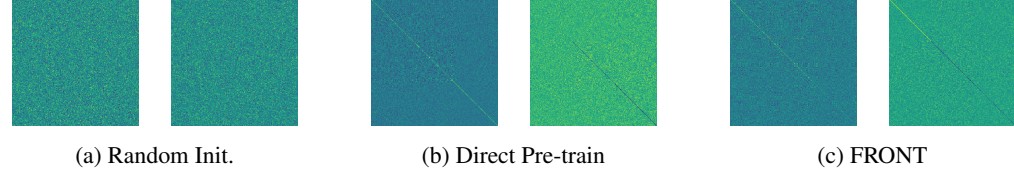

(a) Random Init.          (b) Direct Pre-train          (c) FRONT

Figure 6: Self-attention layer structural patterns illustrated. The figure displays matrices $W_q W_k^T$ (left) and $W_v W_o$ (right) across three DeiT-T configurations: random initialization, pre-trained, and FRONT-initialized models. FRONT can inherit the diagonal property of self-attention layers that only exists in pre-trained ViTs.

As illustrated in Figure 6, FRONT enables models to inherit the essential diagonal attributes within self-attention layers—a characteristic typically exclusive to pre-trained models. FRONT autonomously encapsulates structured knowledge from pre-trained models into the low-frequency domain without requiring manual intervention. Through the transmission of minimal parameters via DCT and IDCT operations, the initialized model inherently preserves these critical structural features in its parameter matrices, resulting in improved training efficiency and faster convergence.

## 5 CONCLUSION

This paper empirically establishes that a model's " learngene "–its core, transferable knowledge–is encoded in the low-frequency components of its weights, providing a robust and efficient mechanism for inheritance by downstream models. We propose FRONT, a novel initialization framework that enables flexible and efficient reuse of parameters from a single pre-trained model to initialize diverse architectures with varying sizes. By leveraging the DCT, FRONT efficiently extracts the task-agnostic, low-frequency knowledge—which we identify as learngenes. Furthermore, we proposed an optional yet powerful frequency-based regularization strategy, FRONT+, which refines learngenes to enhance their generalizability by penalizing high-frequency details. Empirical results demonstrate that our method consistently achieves superior accuracy and significantly reduced training time when initializing models of varying depths and widths, while the size-agnostic representation it leverages proves robust across a range of downstream tasks and datasets.

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

# A PRELIMINARY

## A.1 VISION TRANSFORMER ARCHITECTURE

The Vision Transformer encoder establishes a mapping relationship:

$$\mathcal{E} : \mathbb{R}^{n \times d} \to \mathbb{R}^{n \times d}, \tag{9}$$

composed of $L$ successive transformation layers, where each layer combines multi-head self-attention (MSA) operators with multi-layer perceptron (MLP) projections through residual connections.

The Multi-Head Self-Attention (MSA) mechanism allows the model to attend to different parts of the input sequence, capturing long-range dependencies between image patches. It consists of $h$ parallel self-attention heads. For each attention head $i \in \{1, ..., h\}$, the self-attention head operation computes:

$$\text{Attn}_i(\mathbf{X}) = \text{softmax}\left(\frac{(\mathbf{X}\mathbf{W}_i^q)(\mathbf{X}\mathbf{W}_i^k)^\top}{\sqrt{d_h}}\right)(\mathbf{X}\mathbf{W}_i^v), \tag{10}$$

$$\text{MSA}(\mathbf{X}) = [\text{Attn}_1(\mathbf{X}); \cdots ; \text{Attn}_h(\mathbf{X})]\,\mathbf{W}_o \quad \text{with} \quad \mathbf{W}_o \in \mathbb{R}^{hd_h \times D}, \tag{11}$$

where $\mathbf{W}_i^q, \mathbf{W}_i^k, \mathbf{W}_i^v \in \mathbb{R}^{D \times d_h}$ are learnable projection matrices, and $d_h = D/h$ ensures dimensional consistency. These projections can be compactly represented through concatenation as $\mathbf{W}_{qkv} \in \mathbb{R}^{D \times 3hd}$ where $3hd = 3h \cdot d_h$ corresponds to the combined query-key-value dimensions.

The position-wise MLP implements feature transformation through dimension expansion:

$$\text{MLP}(\mathbf{X}) = \text{GELU}(\mathbf{X}\mathbf{W}_{\text{in}} + \mathbf{b}_1)\mathbf{W}_{\text{out}} + \mathbf{b}_2, \tag{12}$$

where $\mathbf{W}_{\text{in}} \in \mathbb{R}^{D \times D'}$ and $\mathbf{W}_{\text{out}} \in \mathbb{R}^{D' \times D}$ with expansion ratio $\rho = D'/D$ typically set to 4.

The Vision Transformer (ViT) first divides an input image into $N$ patches of size $P \times P$. Each patch is then linearly projected into a $D$-dimensional embedding. To retain positional information, a positional encoding $\mathbf{E} \in \mathbb{R}^{N \times D}$ is added to the patch embeddings, resulting in the input sequence $\mathbf{X} \in \mathbb{R}^{N \times D}$. This sequence is then processed through $L$ successive encoder layers, each implementing dual transformation pathways:

$$\text{EncoderLayer}(\mathbf{X}) = \text{MLP}(\text{LayerNorm}(\text{MSA}(\text{LayerNorm}(\mathbf{X})))) + \mathbf{X}, \tag{13}$$

where MSA denotes multi-head self-attention and MLP represents the position-wise feed-forward network. Layer Normalization (LayerNorm) is applied before both MSA and MLP to stabilize training. This residual structure enables stable gradient flow during backpropagation.

## A.2 DISCRETE COSINE TRANSFORM (DCT) AND INVERSE DCT (IDCT)

**1D-DCT** For $x$ of length $M$, its transformed frequency domain representation $X$ also has a length of $M$:

$$X[k] = \alpha(k) \sum_{n=0}^{M-1} x[n] \cos\left(\frac{\pi(2n+1)k}{2M}\right), \quad k = 0, ..., M-1, \tag{14}$$

where the scaling factor:

$$\alpha(k) = \begin{cases} \sqrt{\dfrac{1}{M}}, & k = 0 \\ \sqrt{\dfrac{2}{M}}, & k \geq 1 \end{cases}. \tag{15}$$

**2D-DCT** For an input matrix $\mathbf{x} \in \mathbb{R}^{M \times N}$, the transformed coefficient matrix $\mathbf{X} \in \mathbb{R}^{M \times N}$ preserves the original dimensions through:

$$X[k,l] = \alpha(k)\alpha(l) \sum_{m=0}^{M-1} \sum_{n=0}^{N-1} x[m,n] \cos\left(\frac{\pi(2m+1)k}{2M}\right) \cos\left(\frac{\pi(2n+1)l}{2N}\right). \quad (16)$$

This is foundational to JPEG compression, where $X[0,0]$ represents the DC (average intensity) component.

**3D-DCT** For the weight $x \in \mathbb{R}^{M \times N \times P}$, the transformed coefficient matrix $X \in \mathbb{R}^{M \times N \times P}$ is:

$$X[k,l,q] = \mathcal{D}(x) = \alpha(k)\alpha(l)\alpha(q) \sum_{m=0}^{M-1} \sum_{n=0}^{N-1} \sum_{p=0}^{P-1} x[m,n,p] \, C(m,n,p,k,l,q). \quad (17)$$

**3D-IDCT** The IDCT recovers the original signal:

$$x[m,n,p] = \mathcal{D}^{-1}(X) = \alpha(k)\alpha(l)\alpha(q) \sum_{k=0}^{M-1} \sum_{l=0}^{N-1} \sum_{q=0}^{P-1} X[k,l,q] \, C(m,n,p,k,l,q), \quad (18)$$

where $0 \le m, k < M$, $0 \le n, l < N$, $0 \le p, q < P$, and $\alpha(d)$ follow the standard definition in 3D-DCT/3D-IDCT:

$$C(m,n,p,k,l,q) = \cos\left(\frac{\pi(2m+1)k}{2M}\right) \cdot \cos\left(\frac{\pi(2n+1)l}{2N}\right) \cdot \cos\left(\frac{\pi(2p+1)q}{2P}\right),$$

$$\alpha(d) = \begin{cases} \sqrt{\frac{1}{S_d}} & \text{if } d = 0 \\ \sqrt{\frac{2}{S_d}} & \text{if } d > 0 \end{cases} \quad \text{with} \quad S_d = \begin{cases} M & \text{if } d = k \\ N & \text{if } d = l \\ P & \text{if } d = q \end{cases}. \quad (19)$$

**4D-DCT** For an input tensor $\mathbf{x} \in \mathbb{R}^{M \times N \times P \times Q}$, the transformed coefficient tensor $\mathbf{X} \in \mathbb{R}^{M \times N \times P \times Q}$ preserves the original dimensions through:

$$X[k,l,p,q] = \alpha(k)\alpha(l)\alpha(p)\alpha(q) \sum_{m=0}^{M-1} \sum_{n=0}^{N-1} \sum_{o=0}^{P-1} \sum_{r=0}^{Q-1} x[m,n,o,r] \cos\left(\frac{\pi(2m+1)k}{2M}\right)$$
$$\cos\left(\frac{\pi(2n+1)l}{2N}\right) \cos\left(\frac{\pi(2o+1)p}{2P}\right) \cos\left(\frac{\pi(2r+1)q}{2Q}\right). \quad (20)$$

**N-Dimensional IDCT** For an input tensor $\mathbf{X} \in \mathbb{R}^{D_1 \times D_2 \times \cdots \times D_N}$, the inverse transformed tensor $\mathbf{x} \in \mathbb{R}^{D_1 \times D_2 \times \cdots \times D_N}$ is given by:

$$x[d_1, d_2, \ldots, d_N] = \prod_{i=1}^{N} \alpha(d_i) \sum_{k_1=0}^{D_1-1} \sum_{k_2=0}^{D_2-1} \cdots \sum_{k_N=0}^{D_N-1} X[k_1, k_2, \ldots, k_N] \prod_{i=1}^{N} \cos\left(\frac{\pi(2d_i+1)k_i}{2D_i}\right), \quad (21)$$

where

$$\alpha(d_i) = \begin{cases} \sqrt{\dfrac{1}{D_i}}, & k_i = 0 \\ \sqrt{\dfrac{2}{D_i}}, & k_i \ge 1 \end{cases}. \quad (22)$$

## B FRONT ALGORITHMS

The following process takes the example of direct extraction (FRONT) and using knowledge distillation to train the auxiliary model from scratch (FRONT+).

## B.1 APPLICATION TO VISION TRANSFORMERS (VITS)

**Algorithm 1: ViTs Learngene Acquisition    Requirements:**

- Pre-trained/Auxiliary model weight tensors $\mathbf{\Omega}_{\text{pre/aux}}$ = $\{\mathbf{W}_{qkv}^{(1 \sim L)}, \mathbf{W}_o^{(1 \sim L)}, \mathbf{W}_{fc1}^{(1 \sim L)}, \mathbf{W}_{fc2}^{(1 \sim L)}\}$, frequency ratio $r$.
- For Method 2: Training dataset $\{(x^{(i)}, y^{(i)})\}_{i=1}^m$, number of epochs $N_{ep}$, batch size $B$, learning rate $\alpha$, regularization weight $\lambda$, decay rates $\gamma_d$.

**Output**: Learngenes (Compact spectral representation).

1. **Method 1: Direct Truncation of pre-trained Weights' Spectra**:
   (a) Initialize an empty set for learngene $\mathcal{G}_1 = \emptyset$.
   (b) For each weight tensor $\mathbf{W}_i \in \mathbf{\Omega}_{\text{pre}}$:
       i. Apply 3D-DCT:
   $$\mathbf{\Phi}_i = \mathcal{D}(\mathbf{W}_i)$$
       ii. Construct binary truncation mask:
   $$\mathbf{M}_r[l, m, n] = \begin{cases} 1 & \text{if } l \leq \lfloor rL \rfloor, m \leq \lfloor rd_{\text{in}} \rfloor, n \leq \lfloor rd_{\text{out}} \rfloor \\ 0 & \text{otherwise} \end{cases} \quad \text{(Eq. (4))}$$
       iii. Apply truncation: $\hat{\mathbf{\Phi}}_i = \mathbf{M}_r \odot \mathbf{\Phi}_i$.
       iv. Add the non-zero elements of $\hat{\mathbf{\Phi}}_i$ to $\mathcal{G}_1$. (Learngenes from Method 1)
   (c) Output $\mathcal{G}_1$ as Learngenes.

2. **Method 2: Distillation-Aware Training with a high-frequency attenuation regularization from scratch**:
   (a) Initialize Auxiliary Model with weights $\mathbf{\Omega}_{\text{aux}}$ (copying $\mathbf{\Omega}_{\text{pre}}$ or random, where we use random Initialization).
   (b) Initialize training optimizer for $\mathbf{\Omega}_{\text{aux}}$.
   (c) For $ep = 1$ to $N_{ep}$:
       i. For each batch $\{(x_j, y_j)\}_{j=1}^B$:
       A. Get logits: $z_{\text{pre},j} = f_{\text{pre}}(x_j)$, $z_{\text{aux},j} = f_{\text{aux}}(x_j)$.
       B. Calculate Logit Loss:
   $$\mathcal{L}_{\text{logit}} = \frac{1}{B} \sum_{j=1}^B (\text{KL}(z_{\text{pre},j} \| z_{\text{aux},j}) + \text{CE}(z_{\text{aux},j}, y_j))$$
       C. Calculate the high-frequency attenuation regularization Loss $\mathcal{L}_{\text{reg}}$:
           • Set $\mathcal{L}_{\text{reg}} = 0$.
           • For each weight tensor $\mathbf{W}_{\text{aux},i} \in \mathbf{\Omega}_{\text{aux}}$:
             ○ Apply 3D-DCT: $\mathbf{\Phi}_{\text{aux},i} = \mathcal{D}(\mathbf{W}_{\text{aux},i})$.
             ○ Construct adaptive mask $\mathbf{M}_r[l, m, n]$ using Eq. (6).
             ○ Calculate term:
   $$Term_i = \frac{1}{\|\mathbf{M}_r \odot \mathbf{\Phi}_{\text{aux},i}\|_0} \sum_{l,m,n} (\mathbf{M}_r[l, m, n] \cdot \mathbf{\Phi}_{\text{aux},i}[l, m, n])^2 \quad \text{(Eq. (5))}$$
             ○ $\mathcal{L}_{\text{reg}} = \mathcal{L}_{\text{reg}} + Term_i$.
       D. Calculate Total Loss:
   $$\mathcal{L}_{\text{total}} = (1 - \lambda)\mathcal{L}_{\text{logit}} + \lambda\mathcal{L}_{\text{reg}} \quad \text{(Eq. (7))}$$
       E. Perform backward pass and update weights:
   $$\mathbf{\Omega}_{\text{aux}} \leftarrow \mathbf{\Omega}_{\text{aux}} - \alpha \nabla \mathcal{L}_{\text{total}}$$
   (d) Initialize $\mathcal{G}_2 = \emptyset$.
   (e) For each trained weight tensor $\mathbf{W}_{\text{aux},i} \in \mathbf{\Omega}_{\text{aux}}$:
       i. Apply 3D-DCT: $\mathbf{\Phi}_{\text{aux},i} = \mathcal{D}(\mathbf{W}_{\text{aux},i})$.
       ii. Add low-frequency components of $\mathbf{\Phi}_{\text{aux},i}$ to $\mathcal{G}_2$. (learngenes from Method 2)
   (f) Output $\mathcal{G}_2$ as learngenes.

**Algorithm 2: ViTs Initialization of Variable-Sized Models** **Input:** Acquired Learngenes $\mathcal{G} = \{\hat{\Phi}_i\}$, target model dimensions $L'$, $d'_{\text{in}}$, $d'_{\text{out}}$.

**Output:** Initialized weight tensors for the variable-sized target model $\Omega' = \{\mathbf{W}_{qkv}^{\prime(1\sim L')}, \mathbf{W}_o^{\prime(1\sim L')}, \mathbf{W}_{fc1}^{\prime(1\sim L')}, \mathbf{W}_{fc2}^{\prime(1\sim L')}\}$.

1. **Spectral Transformation:**

   a. **Case 1:** $L \leq L'$, $d_{\text{in}} \leq d'_{\text{in}}$, $d_{\text{out}} \leq d'_{\text{out}}$ **(Padding):**

      i. Apply zero-padding to $\hat{\Phi}_i$ to obtain $\Phi'_i \in \mathbb{R}^{L' \times d'_{\text{in}} \times d'_{\text{out}}}$. Pad along each dimension as needed.

   b. **Case 2:** $L > L'$, $d_{\text{in}} > d'_{\text{in}}$, $d_{\text{out}} > d'_{\text{out}}$ **(Truncation):**

      i. Truncate $\hat{\Phi}_i$ to obtain $\Phi'_i \in \mathbb{R}^{L' \times d'_{\text{in}} \times d'_{\text{out}}}$. Discard high-frequency elements along each dimension as needed.

   c. **Case 3: Mixed Padding and Truncation:**

      i. Apply padding or truncation along each dimension $(L, d_{\text{in}}, d_{\text{out}})$ as needed to obtain $\Phi'_i \in \mathbb{R}^{L' \times d'_{\text{in}} \times d'_{\text{out}}}$.

2. **Weight Reconstruction:**

   a. Apply IDCT:

   $$\mathbf{W}'_i = \mathcal{D}^{-1}(\Phi'_i) \in \mathbb{R}^{L' \times d'_{\text{in}} \times d'_{\text{out}}}$$

   b. Add the reconstructed weight tensor $\mathbf{W}'_i$ to the set $\Omega'$.

   c. Output $\Omega'$ as the initialized target model weights.

## B.2 APPLICATION TO MULTI-LAYER PERCEPTRONS (MLPs)

The general framework described in Algorithms B.2 and B.2 can be applied to Multi-Layer Perceptrons (MLPs) by adapting the dimensionality of the spectral transformation to match the weight tensors of MLP layers. MLP layers typically have 2D weight matrices $\mathbf{W} \in \mathbb{R}^{D_{\text{in}} \times D_{\text{out}}}$, where $D_{\text{in}}$ and $D_{\text{out}}$ are the input and output dimensions of the layer, respectively. Biases are typically 1D vectors and are handled separately by 1D-DCT/1D-IDCT and not included in the following analysis. It is important to note that arbitrary transformations can be performed on the MLP in terms of width or depth. In the case of width changes as columns, the 2D-DCT/2D-IDCT needs to be applied.

**Algorithm 1: MLP Learngene Acquisition** Applying Algorithm B.2 for Learngene Acquisition to MLPs follows the same two methods, but utilizes the 2D Discrete Cosine Transform ($\mathcal{D}_{2D}$) and its inverse ($\mathcal{D}_{2D}^{-1}$).

**Requirements:**

- Pre-trained/Auxiliary MLP model weight matrices $\Omega_{\text{pre/aux}} = \{\mathbf{W}^{(l)}\}_{l=1}^{L}$, frequency ratio $r$. Here, each $\mathbf{W}^{(l)} \in \mathbb{R}^{D_{\text{in}}^{(l)} \times D_{\text{out}}^{(l)}}$.

- For Method 2: Training dataset $\{(x^{(i)}, y^{(i)})\}_{i=1}^{m}$, number of epochs $N_{ep}$, batch size $B$, learning rate $\alpha$, regularization weight $\lambda$. (Decay rates $\gamma_d$ if applicable).

**Output**: Learngenes (Compact 2D spectral representation).

1. **Method 1: Direct Truncation of pre-trained Weights' Spectra**:

   (a) Initialize an empty set for learngene $\mathcal{G}_1 = \emptyset$.

   (b) For each weight matrix $\mathbf{W}_i \in \Omega_{\text{pre}}$:

      i. Apply 2D-DCT: $\Phi_i = \mathcal{D}_{2D}(\mathbf{W}_i)$.

      ii. Construct binary truncation mask: Adapt Eq. ( 4) for 2D, using indices $(l, m)$ and dimensions $(D_{\text{in},i}, D_{\text{out},i})$ of $\mathbf{W}_i$. The mask $\mathbf{M}_r[l, m]$ is 1 if $l \leq \lfloor r D_{\text{in},i} \rfloor$ and $m \leq \lfloor r D_{\text{out},i} \rfloor$, and 0 otherwise.

      iii. Apply truncation: $\hat{\Phi}_i = \mathbf{M}_r \odot \Phi_i$.

iv. Add the non-zero elements of $\hat{\boldsymbol{\Phi}}_i$ to $\mathcal{G}_1$.

(c) Output $\mathcal{G}_1$ as Learngenes.

2. **Method 2: Distillation-Aware Training with a high-frequency attenuation regularization**:

(a) Initialize Auxiliary MLP Model with weights $\boldsymbol{\Omega}_{\text{aux}}$ (as in Alg. B.1, Method 2 for ViT).

(b) Initialize training optimizer for $\boldsymbol{\Omega}_{\text{aux}}$.

(c) For $ep = 1$ to $N_{ep}$: Steps for batch processing, logit loss $\mathcal{L}_{\text{logit}}$, total loss $\mathcal{L}_{\text{total}}$, and weight updates are analogous to those in Alg. B.1, Method 2 for ViT.

  i. For each batch $\{(x_j, y_j)\}_{j=1}^{B}$:

    A. Get logits $z_{\text{pre},j}, z_{\text{aux},j}$ and calculate $\mathcal{L}_{\text{logit}}$ (as in Alg. B.1, Method 2 for ViT).

    B. Calculate $\mathcal{L}_{\text{reg}}$:

    • Set $\mathcal{L}_{\text{reg}} = 0$.

    • For each weight matrix $\mathbf{W}_{\text{aux},i} \in \boldsymbol{\Omega}_{\text{aux}}$:

      – Apply 2D-DCT: $\boldsymbol{\Phi}_{\text{aux},i} = \mathcal{D}_{2D}(\mathbf{W}_{\text{aux},i})$.

      – Construct adaptive mask $\mathbf{M}_r[l, m]$ using adapted Eq. ( 6) for 2D dimensions.

      – Calculate term: Adapt Eq. ( 5) for 2D dimensions:

$$Term_i = \frac{1}{\|\mathbf{M}_r \odot \boldsymbol{\Phi}_{\text{aux},i}\|_0} \sum_{l,m} (\mathbf{M}_r[l, m] \cdot \boldsymbol{\Phi}_{\text{aux},i}[l, m])^2$$

      – $\mathcal{L}_{\text{reg}} = \mathcal{L}_{\text{reg}} + Term_i$.

    C. Calculate Total Loss $\mathcal{L}_{\text{total}} = (1 - \lambda)\mathcal{L}_{\text{logit}} + \lambda\mathcal{L}_{\text{reg}}$.

    D. Perform backward pass and update weights $\boldsymbol{\Omega}_{\text{aux}}$ (as in Alg. B.1, Method 2 for ViT).

(d) Initialize $\mathcal{G}_2 = \emptyset$.

(e) For each trained weight matrix $\mathbf{W}_{\text{aux},i} \in \boldsymbol{\Omega}_{\text{aux}}$:

  i. Apply 2D-DCT: $\boldsymbol{\Phi}_{\text{aux},i} = \mathcal{D}_{2D}(\mathbf{W}_{\text{aux},i})$.

  ii. Add low-frequency components of $\boldsymbol{\Phi}_{\text{aux},i}$ (as defined by the mask $\mathbf{M}_r$) to $\mathcal{G}_2$.

(f) Output $\mathcal{G}_2$ as learngenes.

**Algorithm 2: MLP Initialization of Variable-Sized Models**  Applying Algorithm B.2 for Initialization to MLPs uses the 2D spectral learngenes to reconstruct weight matrices for a target MLP model with potentially different layer dimensions $D'_{\text{in}}, D'_{\text{out}}$.

**Input:** Acquired MLP Learngenes $\mathcal{G} = \{\hat{\boldsymbol{\Phi}}_i\}$, target MLP layer dimensions $D'_{\text{in}}, D'_{\text{out}}$ for each corresponding layer.

**Output:** Initialized weight matrices for the variable-sized target MLP model $\boldsymbol{\Omega}' = \{\mathbf{W}'^{(l)}\}_{l=1}^{L'}$.

1. **Spectral Transformation:** For each set of learngenes $\hat{\boldsymbol{\Phi}}_i$ corresponding to a layer and its target dimensions $D'_{\text{in}}, D'_{\text{out}}$:

   a. Apply padding or truncation to $\hat{\boldsymbol{\Phi}}_i$ along its two dimensions to obtain $\boldsymbol{\Phi}'_i \in \mathbb{R}^{D'_{\text{in}} \times D'_{\text{out}}}$. The cases for padding/truncation are analogous to those described in Alg. B.1 2 (ViT), but applied to the two dimensions $(D_{\text{in}}, D_{\text{out}})$ of the spectral coefficients.

2. **Weight Reconstruction:**

   a. Apply 2D-IDCT: $\mathbf{W}'_i = \mathcal{D}_{2D}^{-1}(\boldsymbol{\Phi}'_i) \in \mathbb{R}^{D'_{\text{in}} \times D'_{\text{out}}}$.

   b. Add the reconstructed weight matrix $\mathbf{W}'_i$ to the set $\boldsymbol{\Omega}'$.

   c. Output $\boldsymbol{\Omega}'$ as the initialized target MLP model weights.

## B.3   APPLICATION TO CONVOLUTIONAL NEURAL NETWORKS (CNNs)

Applying the FRONT framework to Convolutional Neural Networks (CNNs) follows the same principles, but the spectral transformation must be adapted to the dimensionality of CNN weight tensors. Convolutional layers typically have 4D weight tensors $\mathbf{W} \in \mathbb{R}^{C_{\text{out}} \times C_{\text{in}} \times K_h \times K_w}$, where $C_{\text{out}}$ and $C_{\text{in}}$ are the output and input channels, and $K_h, K_w$ are the kernel dimensions. Biases are usually

1D and handled separately. It is important to note that the four-dimensional tensor of a CNN can be reshaped into a two-dimensional tensor, and subsequently, multiple tensors can be combined into a three-dimensional tensor, and so forth. In order to demonstrate the flexibility of FRONT, the following presentation will outline the method for direct processing of the four-dimensional tensor.

**Algorithm 1: CNN Learngene Acquisition**  Applying Algorithm B.3 for Learngene Acquisition to CNNs utilizes the 4D Discrete Cosine Transform ($\mathcal{D}_{4D}$) and its inverse ($\mathcal{D}_{4D}^{-1}$).

**Requirements:**

- Pre-trained/Auxiliary CNN model weight tensors $\boldsymbol{\Omega}_{\text{pre/aux}} = \{\mathbf{W}^{(l)}\}_{l=1}^{L}$, frequency ratio $r$. Here, each $\mathbf{W}^{(l)} \in \mathbb{R}^{C_{\text{out}}^{(l)} \times C_{\text{in}}^{(l)} \times K_h^{(l)} \times K_w^{(l)}}$.

- For Method 2: Training dataset $\{(x^{(i)}, y^{(i)})\}_{i=1}^{m}$, number of epochs $N_{ep}$, batch size $B$, learning rate $\alpha$, regularization weight $\lambda$. (Decay rates $\gamma_d$ if applicable).

**Output**: Learngenes (Compact 4D spectral representation).

1. **Method 1: Direct Truncation of pre-trained Weights' Spectra**:
   (a) Initialize an empty set for learngene $\mathcal{G}_1 = \emptyset$.
   (b) For each weight tensor $\mathbf{W}_i \in \boldsymbol{\Omega}_{\text{pre}}$:
      i. Apply 4D-DCT: $\boldsymbol{\Phi}_i = \mathcal{D}_{4D}(\mathbf{W}_i)$.
      ii. Construct binary truncation mask: Adapt Eq. ( 4) for 4D, using indices $(c_{\text{out}}, c_{\text{in}}, k_h, k_w)$ and dimensions $(C_{\text{out},i}, C_{\text{in},i}, K_{h,i}, K_{w,i})$ of $\mathbf{W}_i$. The mask $\mathbf{M}_r[c_{\text{out}}, c_{\text{in}}, k_h, k_w]$ is 1 if $c_{\text{out}} \leq \lfloor rC_{\text{out},i} \rfloor$, $c_{\text{in}} \leq \lfloor rC_{\text{in},i} \rfloor$, $k_h \leq \lfloor rK_{h,i} \rfloor$, and $k_w \leq \lfloor rK_{w,i} \rfloor$, and 0 otherwise.
      iii. Apply truncation: $\hat{\boldsymbol{\Phi}}_i = \mathbf{M}_r \odot \boldsymbol{\Phi}_i$.
      iv. Add the non-zero elements of $\hat{\boldsymbol{\Phi}}_i$ to $\mathcal{G}_1$.
   (c) Output $\mathcal{G}_1$ as Learngenes.

2. **Method 2: Distillation-Aware Training with a high-frequency attenuation regularization**:
   (a) Initialize Auxiliary CNN Model with weights $\boldsymbol{\Omega}_{\text{aux}}$ (as in Alg. B.1, Method 2 for ViT).
   (b) Initialize training optimizer for $\boldsymbol{\Omega}_{\text{aux}}$.
   (c) For $ep = 1$ to $N_{ep}$: Steps for batch processing, logit loss $\mathcal{L}_{\text{logit}}$, total loss $\mathcal{L}_{\text{total}}$, and weight updates are analogous to those in Alg. B.1, Method 2 for ViT.
      i. For each batch $\{(x_j, y_j)\}_{j=1}^{B}$:
         A. Get logits $z_{\text{pre},j}, z_{\text{aux},j}$ and calculate $\mathcal{L}_{\text{logit}}$ (as in Alg. B.1, Method 2 for ViT).
         B. Calculate $\mathcal{L}_{\text{reg}}$:
            • Set $\mathcal{L}_{\text{reg}} = 0$.
            • For each weight tensor $\mathbf{W}_{\text{aux},i} \in \boldsymbol{\Omega}_{\text{aux}}$:
               – Apply 4D-DCT: $\boldsymbol{\Phi}_{\text{aux},i} = \mathcal{D}_{4D}(\mathbf{W}_{\text{aux},i})$.
               – Construct adaptive mask $\mathbf{M}_r[c_{\text{out}}, c_{\text{in}}, k_h, k_w]$ using adapted Eq. ( 6) for 4D dimensions.
               – Calculate term: Adapt Eq. ( 5) for 4D dimensions:
               $$Term_i = \frac{1}{\|\mathbf{M}_r \odot \boldsymbol{\Phi}_{\text{aux},i}\|_0} \sum_{c_{\text{out}}, c_{\text{in}}, k_h, k_w} (\mathbf{M}_r[c_{\text{out}}, c_{\text{in}}, k_h, k_w] \cdot \boldsymbol{\Phi}_{\text{aux},i}[c_{\text{out}}, c_{\text{in}}, k_h, k_w])^2$$
               – $\mathcal{L}_{\text{reg}} = \mathcal{L}_{\text{reg}} + Term_i$.
         C. Calculate Total Loss $\mathcal{L}_{\text{total}} = (1 - \lambda)\mathcal{L}_{\text{logit}} + \lambda\mathcal{L}_{\text{reg}}$.
         D. Perform backward pass and update weights $\boldsymbol{\Omega}_{\text{aux}}$ (as in Alg. B.1, Method 2 for ViT).
   (d) Initialize $\mathcal{G}_2 = \emptyset$.
   (e) For each trained weight tensor $\mathbf{W}_{\text{aux},i} \in \boldsymbol{\Omega}_{\text{aux}}$:
      i. Apply 4D-DCT: $\boldsymbol{\Phi}_{\text{aux},i} = \mathcal{D}_{4D}(\mathbf{W}_{\text{aux},i})$.
      ii. Add low-frequency components of $\boldsymbol{\Phi}_{\text{aux},i}$ (as defined by the mask $\mathbf{M}_r$) to $\mathcal{G}_2$.
   (f) Output $\mathcal{G}_2$ as learngenes.

**Algorithm 2: CNN Initialization of Variable-Sized Models**  Applying Algorithm B.3 for Initialization to CNNs uses the 4D spectral learngenes to reconstruct weight tensors for a target CNN model with potentially different layer dimensions $C'_{\text{out}}, C'_{\text{in}}, K'_h, K'_w$.

**Input:** Acquired CNN Learngenes $\mathcal{G} = \{\hat{\mathbf{\Phi}}_i\}$, target CNN layer dimensions $C'_{\text{out}}, C'_{\text{in}}, K'_h, K'_w$ for each corresponding layer.

**Output:** Initialized weight tensors for the variable-sized target CNN model $\mathbf{\Omega}' = \{\mathbf{W}'^{(l)}\}_{l=1}^{L'}$.

1. **Spectral Transformation:** For each set of learngenes $\hat{\mathbf{\Phi}}_i$ corresponding to a layer and its target dimensions $C'_{\text{out}}, C'_{\text{in}}, K'_h, K'_w$:

   a. Apply padding or truncation to $\hat{\mathbf{\Phi}}_i$ along its four dimensions to obtain $\mathbf{\Phi}'_i \in \mathbb{R}^{C'_{\text{out}} \times C'_{\text{in}} \times K'_h \times K'_w}$. The cases for padding/truncation are analogous to those described in Alg. B.1 (ViT), but applied to the four dimensions $(C_{\text{out}}, C_{\text{in}}, K_h, K_w)$ of the spectral coefficients.

2. **Weight Reconstruction:**

   a. Apply 4D-IDCT: $\mathbf{W}'_i = \mathcal{D}_{4D}^{-1}(\mathbf{\Phi}'_i) \in \mathbb{R}^{C'_{\text{out}} \times C'_{\text{in}} \times K'_h \times K'_w}$.
   b. Add the reconstructed weight tensor $\mathbf{W}'_i$ to the set $\mathbf{\Omega}'$.
   c. Output $\mathbf{\Omega}'$ as the initialized target CNN model weights.

## C  TRAINING DETAILS

### C.1  SUPPLEMENTED EXPERIMENTAL SETUP

We systematically evaluate the effectiveness of our method across different architectures and tasks, organizing our analysis by modality. For vision models, we first assess scalability by varying the depth and width of DeiT in Section 4.2.1, including a thorough 300-epoch analysis of direct initialization quality. We then examine cross-dataset and cross-task generalization. For language models, we conduct analogous scaling experiments and evaluate downstream performance on the GLUE benchmark for models like BERT in Section 4.3. Finally, Section 4.4 presents ablation and analyses to understand the underlying mechanisms that make our method effective.

**Datasets.** For vision tasks, all source models, whether used for direct extraction or refinement, are pre-trained on ImageNet-1K. This dataset is also used for our vision scaling experiments. We assess cross-dataset transferability on a diverse suite of seven downstream classification datasets. Furthermore, we evaluate generalization to other vision domains, using six datasets for object detection and four for image segmentation. For language tasks, we conduct training on standard large-scale corpora: the English Wikipedia corpus is used for BERT (Devlin et al., 2019) and RoBERTa (Liu et al., 2019), while the concatenation of English Wikipedia and the Toronto Book Corpus is used for GPT-2 (Liu et al., 2019). The downstream performance of our initialized language models is subsequently evaluated on the GLUE benchmark (Wang et al., 2018).

**Architectural Details.** Our evaluation spans a wide range of architectures in both vision and language domains. For vision models, we utilize publicly available DeiT-Ti/S/B models pre-trained on ImageNet-1K as the source for direct extraction. For our refinement, we train three compact 8-layer auxiliary models (with 3, 6, or 12 heads) from scratch for 150 epochs on ImageNet-1K. We also explore a more efficient refinement variant by briefly fine-tuning the pre-trained DeiT models. To assess scalability, we initialize DeiT variants by varying their depth (4–12 layers) and width (6–24 heads; 384–1536 dimensions). For generalization experiments, we initialize ResNet-50/152 from a ResNet-101 source, and also use ViT-S/B and ResNet-50 as backbones for downstream object detection and semantic segmentation tasks, respectively. For language models, we demonstrate cross-scale transfer by initializing a 6-layer, 384-dimension model (e.g., BERT-S) using learngenes extracted from its corresponding 12-layer, 768-dimension Base counterpart (BERT-B). This procedure is applied across three foundational architectures: BERT, RoBERTa, and GPT-2. For more experimental details, including visual and language models and datasets, please refer to the below appendix.

**Baseline.** To provide a comprehensive evaluation for vision tasks, we benchmark our framework's flexible learngene sourcing strategies against two main categories of initialization methods: **(1) Direct**

Table 8: Hyper-parameters for FRONT's Retraining on ImageNet-1K.

| Training Settings | Configuration |
|---|---|
| optimizer | AdamW |
| base learning rate | Ti: 5e-4 \| S: 2.5e-4 \| B: 1.25e-4 |
| warmup learning rate | 1e-6 |
| weight decay | 0.05 |
| optimizer momentum | 0.9 |
| batch size | Ti: 512 \| S: 256 \| B: 128 |
| training epochs | 150 |
| learning rate schedule | cosine decay |
| warmup epochs | 5 |
| color jitter | 0.4 |
| auto augment | rand-m9-mstd0.5-inc1 |
| mixup | 0.8 |
| cutmix | 1.0 |
| label smoothing | 0.1 |
| drop path | 0.1 |

Table 9: Hyperparameters for neural networks trained on downstream datasets.

| Dataset | Batch Size | Epochs DeiT/ResNet | Learning Rate | Drop Last | Warmup Epochs | Droppath Rate | Color Jitter | Auto Augment | Random Erase | Mixup | Cutmix | Scheduler | Optimizer |
|---|---|---|---|---|---|---|---|---|---|---|---|---|---|
| **Oxford Flowers** | 512 | 300/100 | 3e-4 | False | 0 | 0 | 0.4 | | 0.25 | 0 | 0 | cosine | AdamW |
| **CUB-200-2011** | 512 | 300/100 | 3e-4 | False | 0 | 0.1 | 0 | | 0.25 | 0 | 0 | cosine | AdamW |
| **Stanford Cars** | 512 | 300/100 | 3e-4 | False | 0 | 0.1 | 0 | rand-m9-mstd0.5-inc1 | 0.25 | 0 | 0 | cosine | AdamW |
| **CIFAR10** | 512 | 300/100 | 5e-4 | True | 0 | 0.1 | 0.4 | | 0.25 | 0 | 0 | cosine | AdamW |
| **CIFAR100** | 512 | 300/100 | 5e-4 | True | 0 | 0.1 | 0.4 | | 0.25 | 0 | 0 | cosine | AdamW |
| **Food101** | 512 | 300/100 | 5e-4 | True | 0 | 0.1 | 0.4 | | 0.25 | 0 | 0 | cosine | AdamW |
| **iNat-2019** | 512 | 100/50 | 5e-4 | True | 0 | 0.1 | 0.4 | | 0.25 | 0 | 0 | cosine | AdamW |

**Initialization.** Methods here rely on prior knowledge or direct parameter transfer applied to existing pre-trained models on ImageNet-1K without additional training. This includes He-Init (Chen et al., 2021), Mimetic-Init (Trockman & Kolter, 2023), Wt Select (Xu et al., 2023), Heur-LG (Wang et al., 2022), Cluster-LG (Wang et al., 2024b), LiGO (Wang et al., 2023a), and **FRONT**. **(2) Methods with Extra Training.** This category encompasses methods that require a dedicated optimization phase to generate transferable knowledge. Methods like GHN-3 (Knyazev et al., 2023), Share-Init (Lan et al., 2019), Auto-LG (Wang et al., 2023b), TELG (Xia et al., 2024), and WAVE (Feng et al., 2025b), execute a dedicated, computationally intensive process to generate or discover transferable parameters, a necessity as their frameworks typically do not support the direct fine-tuning of the pre-trained models. To ensure the most direct and rigorous comparison against these high-effort methods, our main experiments also adopt the from-scratch paradigm with **FRONT+**. We use the same pretraining data and experimental settings for all the methods for a fair comparison. Notably, our refinement framework also supports a highly efficient fine-tuning strategy (**FRONT++**). Our preliminary results indicate that this approach can surpass the from-scratch version in performance with significantly fewer training cost, detailed in Section 4.4.1. For language tasks, we established two baselines: training the Small model from scratch and employing knowledge distillation (Hinton et al., 2015), where the respective Base model served as the teacher.

**Evaluation Metrics.** The main metric is Top-1 accuracy, measuring initialization effectiveness. Supplementary metrics are convergence efficiency (epochs) and parameter transfer efficiency.

## C.2 VISION TASKS

### C.2.1 HYPER-PARAMETERS

Table 8 and Table 9 detail the hyperparameters used for FRONT+, which concentrates knowledge into low frequencies, and for training models initialized with this low-frequency knowledge on various datasets. These hyperparameters include batch sizes, warm-up epochs, training epochs, and other relevant settings.

### C.2.2 DETAILS OF FRONT'S DIRECT SETTING

One-time extraction with primary hyperparameter being extracted parameter quantity.

Table 10: Characteristics of downstream datasets.

| Dataset | Classes | Total | Training | Testing |
|---|---|---|---|---|
| **Oxford Flowers** (nil, 2008) | 102 | 8,189 | 2,040 | 6,149 |
| **CUB-200-2011** (Wah et al., 2011) | 200 | 11,788 | 5,994 | 5,794 |
| **Stanford Cars** (Gebru et al., 2017) | 196 | 16,185 | 8,144 | 8,041 |
| **CIFAR10** (Krizhevsky & Hinton, 2009) | 10 | 60,000 | 50,000 | 10,000 |
| **CIFAR100** (Krizhevsky & Hinton, 2009) | 100 | 60,000 | 50,000 | 10,000 |
| **Food101** (Bossard et al., 2014) | 101 | 101,000 | 75,750 | 25,250 |
| **iNat-2019** (Tan et al., 2019) | 1010 | 268,243 | 265,213 | 3,030 |

**Depth Expansion**: For tiny models, search space is $\{1.7, 2.2, 2.7\}$M parameters, with 2.2M selected as optimal. For small models, search space is $\{8.1, 11.4\}$M parameters, with 8.1M selected as optimal. For base models, search space is $\{32.4, 46.3\}$M parameters, with 32.4M selected as optimal.

**Width Expansion**: We uniformly use a parameter size of 8.9M for transmission.

**Downstream Tasks**: In the settings of Table 3, for the tiny model, we initialize the 6-layer tiny model using a publicly available 12-layer pre-trained tiny model, transferring 2.2M parameters. Similarly, for the small model, the 6-layer small model is initialized with a 12-layer publicly available pre-trained small model, transferring 8.1M parameters. For object detection, we initialize the 12-layer ViT-S model (with an embedding dimension of 384) using weights from a pre-trained 12-layer ViT-B model (dimension 768), and initialize the ViT-B model from a pre-trained ViT-L model (dimension 1024). For image segmentation, we initialize the ResNet50 model with weights from a pre-trained ResNet101 model.

### C.2.3 DETAILS OF FRONT+'S REFINEMENT

Our Refinement strategy is implemented via two distinct approaches:

**Training from Scratch (FRONT+).** For our main experiments requiring a from-scratch model, we train compact auxiliary networks for 150 epochs on ImageNet-1K. To enhance their representational quality, we employ knowledge distillation, using either a LeViT-384 or a RegNetY-16GF model as the teacher. The learngenes are then directly extracted from these fully trained auxiliary models. A detailed ablation study on the choice of teacher models can be found in Appendix D.4.

Hyperparameters include regularization coefficient in Eq. (7), transferred parameter quantity and learning rate. We maintain consistent parameter quantities for fair comparison: 0.8M for tiny models, 3.2M for small models, 13.0M for base models. Regularization coefficient searched over $\{0.5, 0.2, 0.1, 0.05, 0.002, 0.001, 0.0001\}$, with 0.002 selected based on convergence and regularization effectiveness. We select the learning rate for search in $\{1e-3, 5e-4\}$. During the training of the auxiliary model, we use $l = 2$ and set $m, n = 0.25 \times d_{\text{in}}, d_{\text{out}}$ in the mask matrix, where $d_{\text{in}}$ and $d_{\text{out}}$ denote the input and output dimension, respectively.

**Fine-tuning a Pre-trained Model (FRONT++).** To provide a more computationally efficient alternative, this approach initializes with a publicly available pre-trained model and then briefly fine-tunes it with our frequency regularization term. As detailed in our ablation analysis (Section 4.4.1), we conduct a grid search to determine the optimal configuration. Specifically, a DeiT-Tiny-L8 source model is fine-tuned on ImageNet-1K across a range of epochs $\{10, 20, 50, 100\}$ and regularization strengths ($\lambda$) $\{1e-5, 5e-4, 1e-3, 2e-3, 5e-3\}$. The quality of the learngenes (0.8M parameters) produced by each configuration is then assessed by their effectiveness in initializing a deeper DeiT-Tiny-L10 target model.

### C.2.4 DETAILS OF DOWNSTREAM DATASETS

Additional datasets include Oxford Flowers (nil, 2008), CUB-200-2011 (Wah et al., 2011), Stanford Cars (Gebru et al., 2017), CIFAR-10, CIFAR-100 (Krizhevsky & Hinton, 2009), Food-101 (Bossard et al., 2014), and iNaturalist-2019 (Tan et al., 2019). Table 10 presents the details of seven downstream datasets, which are sorted by the size of datasets.

Following the setting (Nie et al., 2024), it is clear that FRONT effectively transfers knowledge acquired in the source domain to the four target domains compared to from scratch. On DeepGlobe (Demir et al., 2018), FRONT achieves comparable accuracy when segmenting various categories from

satellite imagery. Furthermore, FRONT produces precise segmentations for medical screening data from ISIC (Codella et al., 2019) and Chest X-Ray (Tschandl et al., 2018). Regarding FSS-1000 (Li et al., 2020b), FRONT delivers strong performance when predicting a wide range of target categories, such as logos, food items, and other objects.

Following the setting (Fu et al., 2024), COCO (Lin et al., 2014) is a widely adopted dataset for object detection, offering a broad range of categories including humans, animals, vehicles, and common items; it is utilized as the source domain dataset. The remaining six datasets—ArTaxOr (Drange, 2019), Clipart1k (Inoue et al., 2018), DIOR (Li et al., 2020a), DeepFish (Saleh et al., 2020), NEUDET (Song & Yan, 2013), and UODD (Jiang et al., 2021)—are employed as target domain datasets.

Table 11: Comparison of different basis transforms for knowledge transfer on ImageNet-1K. Our DCT-based method (FRONT) achieves the **highest** top-1 accuracy (%) in both depth and width transfer tasks. This validates our design choice.

| Methods | Scratch | **FRONT** | DFT | DWT | PCA | SVD |
|---|---|---|---|---|---|---|
| DeiT-Ti_L12 → DeiT-Ti_L6 | 40.6 | **63.3** | 60.8 | 62.0 | 57.9 | 53.0 |
| DeiT-S_L6 → DeiT-Ti_L6 | 40.6 | **56.9** | 47.9 | 55.4 | 50.2 | 43.6 |

### C.3  LANGUAGE TASKS

To evaluate the effectiveness and adaptability of our proposed method, we conducted comparative experiments on language models. We applied FRONT to initialize a 6-layer, 384-dimensional "Small" model (e.g., BERT-S) from its corresponding pre-trained 12-layer, 768-dimensional "Base" (e.g., BERT-B) model. This procedure was performed for three distinct architectures: BERT, RoBERTa, and GPT-2. For comparison, we established two baselines: training the Small model from scratch and employing knowledge distillation (Hinton et al., 2015), where the respective Base model served as the teacher. We use the English Wikipedia corpus for training BERT (Devlin et al., 2019) and RoBERTa (Liu et al., 2019). We use the concatenation of English Wikipedia and Toronto Book Corpus for training GPT2 (Radford et al., 2019). We remove the next sentence prediction task ( (Liu et al., 2019)) and use a fixed sequence length of 128 for pretraining both BERT and RoBERTa. For BERT, we use a batch size of 256 and a learning rate of 2e-4, while we use a batch size of 1024 and a learning rate of 8e-4 for training RoBERTa models. For GPT2, we use a batch size of 512, a fixed sequence length of 1024 and a learning rate of 1e-3.

Subsequently, we evaluated the downstream performance of the initialized BERT-S model on the GLUE benchmark. For fine-tuning, we used the Adam optimizer with a learning rate of 2e-5, a batch size of 32, and searched for the optimal number of epochs from {3, 6, 10}. The FRONT-initialized models were directly fine-tuned on each task and knowledge distillation requires the corresponding base models to be used as teacher models throughout the entire training process.

## D  ADDITIONAL RESULTS

### D.1  ABLATION STUDY ON TRANSFORM BASIS CHOICE

Regarding our choice of the DCT over other transforms, we provide this section with a concise theoretical justification followed by a comprehensive empirical ablation study.

Our primary motivation for using DCT is that "low-frequency" in the DCT domain in the signal processing has a universal, and semantically meaningful interpretation: **smoothness**. The cosine basis functions are fixed *a priori*. This allows us to posit a testable hypothesis: that the generalizable, task-agnostic knowledge in weights is encoded in these universal patterns. Furthermore, DCT is a **real-to-real** transform, which is perfectly compatible with real-valued neural network weights.

Our theoretical analysis of the alternatives is as follows:

- **DFT (Fourier):** DFT's primary drawback is that it is a **complex-valued** transform. It maps real-valued weights to complex coefficients (magnitude and phase), forcing a non-trivial

choice: either discard the phase (information loss) or double the parameter count. DCT avoids this ambiguity.

- **DWT (Wavelets):** DWT's strength is its excellent **spatial localization** (analyzing *where* a frequency occurs). For our task, this localization is unnecessary. Our hypothesis is about the **global spectral properties** of the entire weight tensor, not the local position of specific patterns.

- **PCA/SVD (Intra-Model Compression):** This is the key distinction. PCA and SVD are powerful tools for *intra-model compression*, not *cross-model initialization*. It is ill-defined how to use the components (PCA) or singular vectors (SVD) from a source model's weight matrix (e.g., $1024 \times 512$) to initialize a target model's matrix (e.g., $512 \times 256$) without destroying the structural knowledge we aim to transfer.

Therefore, DCT is uniquely suited as it is a **structure-preserving transform**, not a dimensionality reduction or decomposition technique.

While the theoretical arguments above guided our design, we agree that empirical validation is crucial. We conducted a comprehensive ablation study comparing our DCT-based method (FRONT) against baselines using DFT, DWT, PCA, and SVD. We performed experiments for both depth and width transformations. **Depth Transfer:** Using a 12-layer DeiT-Ti to initialize a 6-layer DeiT-Ti. **Width Transfer:** Using a 6-layer DeiT-S to initialize a 6-layer DeiT-Ti. All other training parameters are kept consistent with the settings in the main paper's Table 1. The results are shown in Table 11.

The empirical results in Table 11 strongly validate our design choices. Our DCT-based method, FRONT, clearly outperforms all other baselines, leveraging its energy compaction property. As hypothesized, the frequency-domain methods (DCT, DWT, DFT) are significantly superior to intra-model compression tools (PCA, SVD) for knowledge transfer, confirming our theoretical concerns about the inapplicability of the latter. Notably, while depth transfer (63.3%) is highly effective, the performance drop in width transfer (56.9%)—despite comparable source model performance—highlights the significant challenge posed by transformations that disrupt dimensional structure. Finally, the strong showing of DWT (achieving the second-best results) suggests it is a promising avenue for future research in this area.

Table 12: Performance comparison on ResNet50 and ResNet152.

| | ResNet50 | | ResNet152 | |
|---|---|---|---|---|
| | Rand. | FRONT | Rand. | FRONT |
| Image. | 74.1 | 76.1 ↑ 2.0 | 76.8 | 77.2 ↑ 0.4 |
| Flow. | 51.3 | 91.3 ↑ 40.0 | 28.0 | 87.7 ↑ 59.7 |
| CUB | 40.7 | 68.9 ↑ 28.2 | 22.8 | 68.2 ↑ 45.4 |
| Cars | 48.6 | 88.2 ↑ 39.6 | 29.3 | 87.4 ↑ 58.1 |
| C10 | 94.8 | 95.6 ↑ 0.8 | 95.6 | 95.7 ↑ 0.1 |
| C100 | 76.8 | 78.3 ↑ 1.5 | 77.7 | 79.5 ↑ 1.8 |
| Food | 82.1 | 85.8 ↑ 3.7 | 83.2 | 85.9 ↑ 2.7 |
| iNat | 61.8 | 69.0 ↑ 7.2 | 62.7 | 69.1 ↑ 6.4 |

## D.2 INITIALIZATION ON RESNET-BASED ARCHITECTURES.

To further assess the generality of FRONT, we conduct experiments on CNNs using ResNet models. A community-trained ResNet-101 on ImageNet-1K is used to initialize both ResNet-50 and ResNet-152, which are then evaluated on seven downstream tasks directly and compared against randomly initialized baselines. Results are shown in Table 12.

FRONT consistently outperforms random initialization across all datasets, with particularly notable improvements on fine-grained, small-sample datasets. For example, on the Flowers dataset, FRONT boosts accuracy by 40.0% (ResNet-50) and 59.7% (ResNet-152); on CUB, by 28.2% and 45.4%; and on Stanford Cars, by 39.6% and 58.1%, respectively. These results highlight FRONT's ability to address the difficulty of training deep networks on limited data by leveraging pre-trained models. FRONT thus provides more effective initialization and demonstrates strong generalization, enabling substantial gains even on datasets with unique characteristics and CNN-based architectures.

Table 13: Low-frequency knowledge surpasses others.

| Select Freq. | DeiT-Ti | | | | |
|---|---|---|---|---|---|
| | L4 | L6 | L8 | L10 | L12 |
| He-Init | 34.7 | 40.6 | 43.7 | 46.8 | 48.3 |
| High | 32.6 | 36.0 | 35.8 | 38.9 | 39.0 |
| Mid | 34.0 | 34.6 | 37.0 | 38.1 | 38.2 |
| Low | **48.6** | **53.7** | **56.5** | **58.6** | **59.6** |

## D.3 GENERALIZATION OF LOW-FREQUENCY COMPONENTS

To systematically evaluate the contribution of different frequency components to knowledge transfer, we initialize DeiT-Ti(192 hidden dimensions) using frequency-specific components extracted from

each layer of DeiT-S(384 hidden dimensions). Given $x \in \{4, 6, 8, 10, 12\}$, representing the number of layers considered, and for each layer $i$ where $0 \leq i < x$, we partition the weight matrix $W\text{-}S_i$ into low-frequency ($W\text{-}S_i[: 192, : 192]$), mid-frequency ($W\text{-}S_i[96 : 288, 96 : 288]$), and high-frequency ($W\text{-}S_i[192 :, 192 :]$) components. As shown in Table 13, models initialized with low-frequency components consistently achieve the highest performance, highlighting the superior transferability.

## D.4   INTEGRATION OF KNOWLEDGE FROM LARGER PRE-TRAINED MODELS

Table 14: Additional results on different teacher models.

|  |  | DeiT-Ti | | | DeiT-S | | |
|---|---|---|---|---|---|---|---|
|  | Ancestry | $L_6$ | $L_8$ | $L_{12}$ | $L_4$ | $L_8$ | $L_{12}$ |
| TLEG (Xia et al., 2024) | LeVit-384 (39.1M) | 60.5 | 62.9 | 65.4 | 70.5 | 72.1 | 73.8 |
| WAVE (Feng et al., 2025b) | LeVit-384 (39.1M) | 63.2 | 65.4 | 67.3 | 72.7 | 74.1 | 75.3 |
| FRONT+ | LeVit-384 (39.1M) | **63.4** | **65.6** | **67.5** | **72.9** | **74.2** | **75.4** |
| FRONT+ | RegNet-16GF (83.6M) | 63.3* | 65.2* | **67.5** | 72.4* | **74.2** | 75.3* |

Low-frequency knowledge can be integrated with relevant information extracted from pre-trained models by filtering out size-specific components that violate transformation constraints. This is achieved by zero-padding or truncation in DCT / IDCT process. Such a mechanism ensures the efficient transfer and sharing of size-agnostic knowledge across models of varying sizes.

To evaluate the influence of pre-trained teacher models with varying architectures and scales, we incorporated a larger pre-trained model, RegNet-16GF (83.6M parameters) (Radosavovic et al., 2020), and compared FRONT+ to WAVE and TLEG, both of which utilize LeVit-384 (39.1M parameters) as an alternative teacher model. Table 14 presents the results for DeiT-Ti and DeiT-S at each layer.

The results demonstrate that FRONT+ consistently outperforms the other methods across all model sizes. Although the inclusion of a larger teacher model (RegNet-16GF) provides some improvement, the gains are relatively modest. This suggests that once pre-trained models are sufficiently trained and informative, the shared low-frequency knowledge becomes effective and robust to further increases in teacher model size. These findings highlight the effectiveness and stability of FRONT+ in condensing and integrating low-frequency knowledge from various retrained pre-trained models.

## D.5   EFFECT OF SOURCE PRE-TRAINED MODEL SCALE FOR LEARNGENES

The scale of pre-trained models significantly influences initialization effectiveness. Our results indicate that initializing with pre-trained models of similar or smaller scale produces superior outcomes. Table 15 presents a comparison of using DeiT-Ti, DeiT-S, and DeiT-B weights to directly initialize DeiT-Ti on ImageNet-1K for 10 epochs without additional training. Notably, even a 2.2M parameter subset extracted from the 78M parameters of DeiT-B provides effective initialization, resulting in a 2.2% average accuracy improvement.

Table 15: Pre-trained model's scale. Pre-trained models who are more similar provide better initialization.

| PT. Model | DeiT-Ti | | | | |
|---|---|---|---|---|---|
|  | L4 | L6 | L8 | L10 | L12 |
| He-Init | 34.7 | 40.6 | 43.7 | 46.8 | 48.3 |
| DeiT-B | 29.3 | 39.2 | 47.7 | 51.8 | 57.1 |
| DeiT-S | 34.8 | 45.5 | 51.9 | 56.7 | 59.7 |
| DeiT-Ti | **55.3** | **63.3** | **64.4** | **64.7** | **65.3** |

## D.6   FREQUENCY-DEPENDENT ADAPTATION PATTERNS IN FRONT-INITIALIZED MODELS

Our analysis reveals distinct patterns in how FRONT-initialized models adapt to downstream tasks during fine-tuning. Figure 7 presents similarity scores between weights across transformer components fine-tuned on various downstream datasets detailed in Section C.2.4. Figure 8 reveals that publicly available pre-trained DeiT models exhibit different frequency-domain characteristics compared to randomly initialized models.

Low-frequency parameter components maintain consistently high similarity scores across modules and tasks, indicating these foundational elements remain relatively effective during fine-tuning. In contrast, high-frequency components display significantly lower similarity scores, suggesting substantial updates during adaptation. This pattern supports our hypothesis that low-frequency

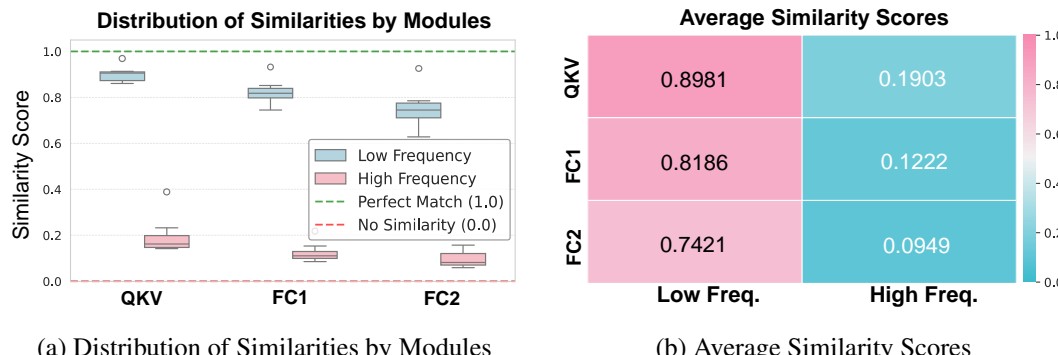

(a) Distribution of Similarities by Modules

(b) Average Similarity Scores

Figure 7: Analysis of Module-wise Similarity Distribution and Average Scores for QKV, FC1, and FC2 Across Downstream Tasks

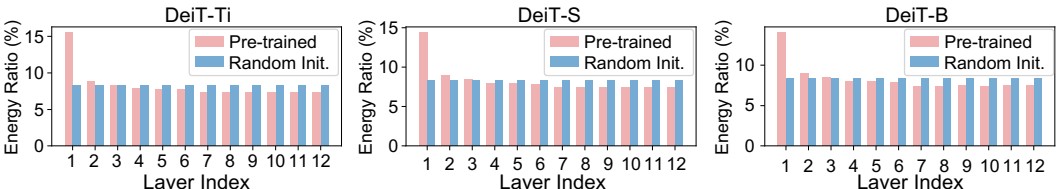

Figure 8: Freqency domain energy distribution comparison between pre-trained and randomly initialized DeiT-Ti/DeiT-S/DeiT-B models after 3D-DCT.

components encode generalized linguistic knowledge transferable across tasks, while high-frequency parameters specialize to capture task-specific nuances. The model appears to selectively update high-frequency components while preserving low-frequency elements during adaptation. These findings provide valuable insights for developing more efficient fine-tuning strategies and validate the theoretical underpinnings of our approach.

### D.7 FASTER CONVERGENCE AND STRONGER LEARNING ABILITY

Figure 9 demonstrates the efficacy of various learngene methods in initializing DeiT models across different scales (tiny, small, base) and layer configurations (L4-L12). Results indicate that FRONT exhibits superior initialization capabilities compared to alternative learngene approaches and performs comparably to models trained from scratch for 150 epochs.

FRONT consistently achieved higher accuracy during the initial 10 epochs across most models. Notably, FRONT does not require the introduction of stochastic parameters, suggesting its effectiveness derives from a deterministic, efficient, and comprehensive initialization strategy. All models demonstrated robust initialization performance in the first training epoch, with FRONT achieving an average accuracy of 70.08% by epoch 7—merely 0.12 percentage points below the benchmark value of 70.20% obtained after 150 epochs of scratch training. This initialization capability effectively compresses the model warm-up phase by approximately 21-fold, substantially reducing computational costs and highlighting FRONT's potential to accelerate deep learning workflows. This represents a significant advantage over certain learngene methods that necessitate additional parameters or complex architectural modifications. FRONT and WAVE typically exhibit similar performance patterns, generally outperforming TLEG, Auto LG, and Heru-LG during early epochs. While Heru LG demonstrates relatively lower accuracy in initial training stages, its performance improves with continued training. As an efficient parameter initialization method, FRONT significantly accelerates downstream model training while maintaining final performance comparable to fully trained baseline models.

### D.8 CROSS-ARCHITECTURE INITIALIZATION

Following the constructive feedback from our reviewers, we conducted two new sets of experiments to validate our framework's ability to handle *heteromorphic* (cross-architecture) transfers, in addition to

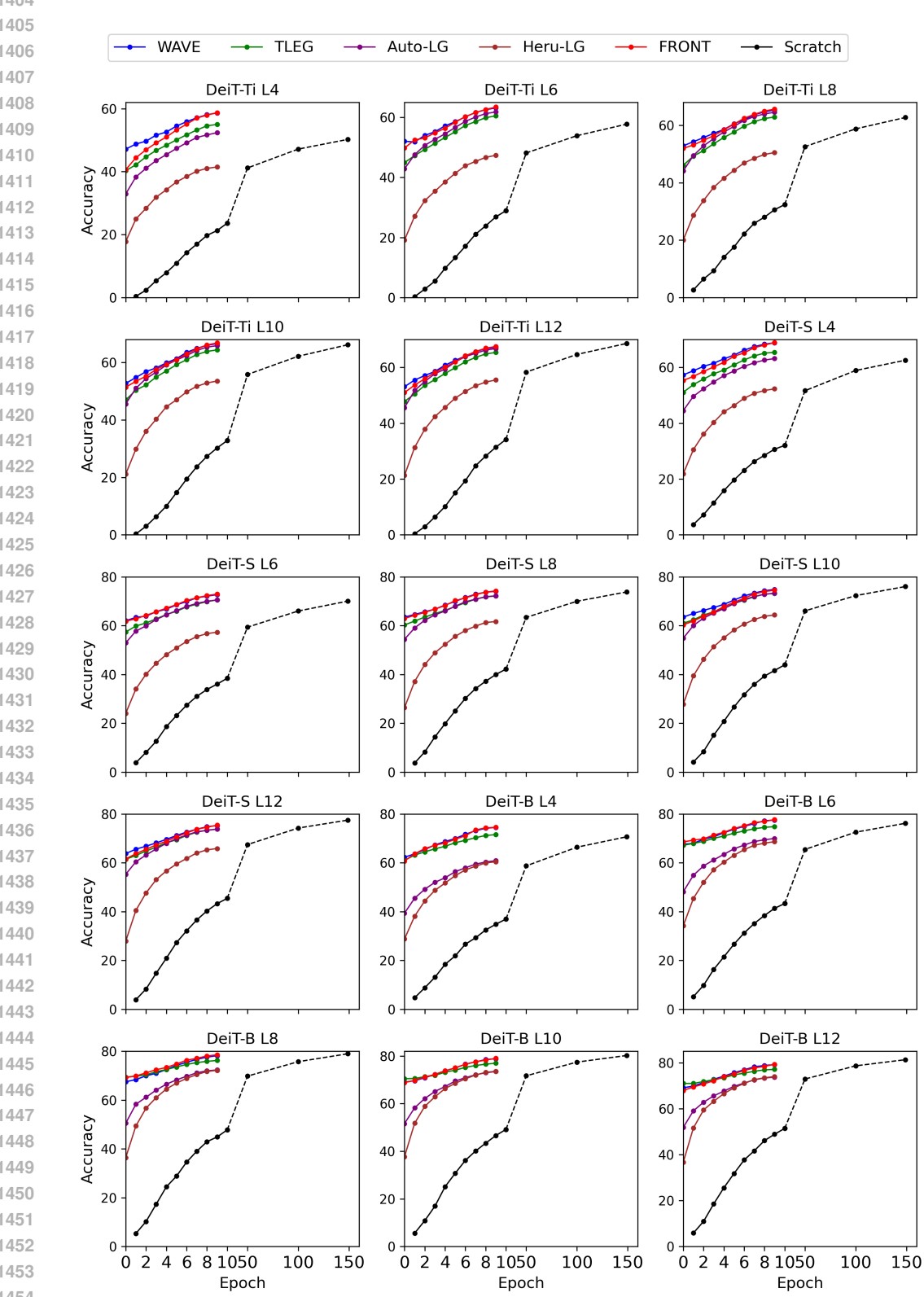

Figure 9: Performance comparisons on ImageNet-1K about depth expansion among FRONT and other learngene methods.

the *homomorphic* (same-family) transfers in the main paper. These experiments involve simultaneous changes in both architecture and model size.

**Language Models: BERT (Encoder) ↔ GPT (Decoder).** We performed a challenging bidirectional transfer between two foundation models with fundamentally different architectures and pre-training objectives: BERT (an Encoder-only model trained on Masked Language Modeling) and GPT (a Decoder-only model trained on Causal Language Modeling / next-token prediction). We transferred knowledge from 12-layer Base models (BERT-B, GPT-B) to 6-layer Small models (BERT-S, GPT-S).

As shown in Figure 10, the transfer was highly successful.

- **GPT-B (L12) → BERT-S (L6)** (Figure 10a): Our method saves **15.4%** of the pre-training FLOPs required to reach the target loss, compared to training BERT-S from Scratch.
- **BERT-B (L12) → GPT-S (L6)** (Figure 10b): This direction showed even more remarkable results, saving **27.0%** of the pre-training FLOPs. We hypothesize this is because the Masked Language Model objective of BERT implicitly contains the next-token prediction capability, and our method successfully extracts and transfers this shared, fundamental knowledge.

These results support our hypothesis that low-frequency spectral components capture task-agnostic functions, enabling effective initialization even when attention mechanisms or encoder/decoder paradigms differ.

**Vision Models: Standard → Parallel Attention.** To address the question of "fundamentally different architectures" in vision, we transferred knowledge from a standard Transformer (DeiT-B L12) to a model using a *parallel attention* block (Mega-ViT-S L6) Dehghani et al. (2023). This represents both an architectural and size mismatch. As shown in Figure 10c, our method (FRONT) provides a substantial initialization boost. Compared to training from Scratch, our initialized model saves **33.6%** of the computational cost (FLOPs) required to reach the target accuracy on the Cifar10 dataset. Parallel attention modifies the mapping between query/key/value interactions, making spectral preservation across depth and width dimensions more demanding than homomorphic transfers.

These experiments demonstrate that our framework is a robust method for decoupling and transferring foundational knowledge, even between models with significant architectural disparities.

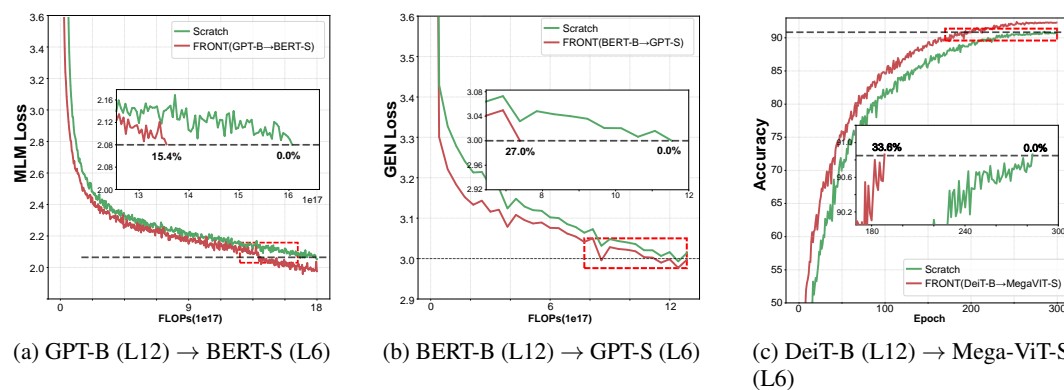

(a) GPT-B (L12) → BERT-S (L6)   (b) BERT-B (L12) → GPT-S (L6)   (c) DeiT-B (L12) → Mega-ViT-S (L6)

Figure 10: Results of cross-architecture and cross-size initialization. **(a)** FRONT(GPT-B→BERT-S) saves **15.4%** FLOPs. **(b)** FRONT(BERT-B→GPT-S) saves **27.0%** FLOPs. **(c)** FRONT(DeiT-B→Mega-ViT-S) saves **33.6%** computational cost. All results are compared to training from Scratch.

## E  THEORETICAL ANALYSIS

In this section, we provide a formal analysis for the claim that low-frequency components of model weights encode task-invariant knowledge. We formulate the fine-tuning process as learning a target function composed of a shared, smooth component and a specific, high-frequency component. We prove that under standard spectral bias assumptions, the optimization gradients are concentrated in the high-frequency weight subspace.

Let $f(x; W) : \mathcal{X} \to \mathcal{Y}$ be a neural network parameterized by $W \in \mathbb{R}^d$. Let $\hat{W} = \mathcal{D}W$. We partition the frequency spectrum $\mathcal{K}$ into low-frequency indices $\mathcal{I}_L$ and high-frequency indices $\mathcal{I}_H$. The weight space $\mathbb{R}^d$ is the direct sum of two orthogonal subspaces $\mathcal{W}_L \oplus \mathcal{W}_H$, where:

$$W_L = \mathcal{D}^{-1}(M_L \odot \hat{W}) \in \mathcal{W}_L, \quad W_H = \mathcal{D}^{-1}(M_H \odot \hat{W}) \in \mathcal{W}_H$$

. Here, $M_L$ and $M_H$ are binary masks. Due to the orthogonality of DCT, $\|W\|^2 = \|W_L\|^2 + \|W_H\|^2$.

**Assumption E.1.** The target function $f^*$ for a downstream task can be decomposed into a shared, generalizable component $f_G^*$ and a task-specific component $f_S^*$:

$$f^*(x) = f_G^*(x) + f_S^*(x),$$

where $f_G^*$ captures common general knowledge and $f_S^*$ captures task-specific details.

**Assumption E.2.** There exists a correspondence between the frequency spectrum of the weights and the Lipschitz smoothness of the function. Specifically, $f_G^*$ is $\lambda_L$-Lipschitz (smooth) and can be well-approximated by weights in $\mathcal{W}_L$ and $f_S^*$ is $\lambda_H$-Lipschitz (sharp), where $\lambda_H \gg \lambda_L$, requiring weights in $\mathcal{W}_H$ for accurate approximation.

**Assumption E.3.** The pre-trained weights $W_0$ have converged to the generalizable component. That is:

$$f(x; W_0) \approx f_G^*(x).$$

Consequently, the projection of $W_0$ onto the generalizable manifold is optimal, implying $\nabla_{W_L} \mathcal{L}_{task}(W_0) \approx 0$ if the task were purely $f_G^*$.

We then analyze the gradient when fine-tuning on the target task with loss function $\mathcal{L}(W) = \frac{1}{2}\mathbb{E}_x[\|f(x; W) - f^*(x)\|^2]$.

**Theorem E.4.** *Under Assumptions E.1-E.3, the gradient of the loss function at the onset of fine-tuning, $G = \nabla_W \mathcal{L}(W_0)$, satisfies:*

$$\|\mathcal{P}_H(G)\| \gg \|\mathcal{P}_L(G)\|,$$

*where $\mathcal{P}_H$ and $\mathcal{P}_L$ are orthogonal projections onto the high-frequency ($\mathcal{W}_H$) and low-frequency ($\mathcal{W}_L$) subspaces, respectively.*

*Proof.* The gradient of the loss $\mathcal{L}(W)$ with respect to weights $W$ is given by:

$$G = \nabla_W \mathcal{L}(W_0) = \mathbb{E}_x \left[ \nabla_W f(x; W_0)^T \left( f(x; W_0) - f^*(x) \right) \right]$$

Using Assumption E.3 and Assumption E.1, the residual error term becomes:

$$r(x) = f(x; W_0) - f^*(x) \approx f_G^*(x) - (f_G^*(x) + f_S^*(x)) = -f_S^*(x).$$

Thus, the optimization is driven solely by the negative task-specific component $-f_S^*(x)$. We project the gradient $G$ onto the low-frequency subspace $\mathcal{W}_L$:

$$\mathcal{P}_L(G) = \mathbb{E}_x \left[ \mathcal{P}_L \left( \nabla_W f(x; W_0) \right)^T \left( -f_S^*(x) \right) \right]$$

Here, $\nabla_W f(x; W_0)$ represents the Jacobian of the network. The term $\mathcal{P}_L(\nabla_W f)$ captures the sensitivity of the network output to changes in low-frequency weights. Based on Assumption E.2, functions generated by variations in $\mathcal{W}_L$ are smooth. Let $\phi_L(x)$ be a basis function in the function space spanned by varying $W_L$. Conversely, $f_S^*(x)$ is a high-frequency (sharp) function. In the frequency domain of the function space, the inner product between a smooth function (variations caused by $W_L$) and a high-frequency residual ($f_S^*$) is negligible due to spectral separation, i.e., $\langle \text{Smooth}, \text{Sharp} \rangle \approx 0$. Therefore:

$$\|\mathcal{P}_L(G)\| = \|\mathbb{E}_x[\nabla_{W_L} f \cdot (-f_S^*)]\| \approx 0.$$

This result aligns with the intuition that low-frequency weights cannot effectively fit high-frequency noise/details; thus, the gradient (which represents the "desire" to fit the error) is small in this direction. Consider the projection onto $\mathcal{W}_H$:

$$\mathcal{P}_H(G) = \mathbb{E}_x \left[ \nabla_{W_H} f(x; W_0)^T (-f_S^*(x)) \right].$$

Since variations in $W_H$ can generate high-frequency functions, the spectral support of $\nabla_{W_H} f$ overlaps significantly with that of $f_S^*$. The inner product is non-zero and significant:

$$\|\mathcal{P}_H(G)\| \propto \|f_S^*\| \gg 0.$$

Then we have: $\|\mathcal{P}_H(G)\| \gg \|\mathcal{P}_L(G)\|$, which concludes the proof.

$\square$

Since the gradient for low-frequency components is negligible, these weights remain stable during fine-tuning, confirming their role as the immutable "learngene". In contrast, high-frequency weights require significant updates to fit $f_S^*$. FRONT explicitly preserves the stable $W_L$ while allowing $W_H$ to be re-initialized or adapted, thereby aligning the initialization with the spectral requirements of transfer learning.

## F  LIMITATIONS AND FUTURE WORK

Our method exhibits limitations when the architectural gap between source and target models is extreme. For example, transferring from a self-attention–dominant network (e.g., Transformer) to a convolution-only network (e.g., pure CNN) typically degrades performance, as these architectures lack the shared tensor-axis semantics required by our DCT-based learner representation (Assumption 1 in Appendix E). These cross-paradigm transfers represent known difficulties in the field and merit systematic investigation.

Future work should focus on developing more general transformation techniques for pre-trained models. Specifically, we aim to:

- Optimize extraction processes to better capture activated parameters from pre-trained weights.
- Design transformation approaches that maintain accuracy without additional training cost.
- Extend the method to handle extreme cross-paradigm scenarios by learning intermediate representations robust to axis misalignment.
- Explore adaptive mechanisms for high-frequency (HF) component transfer, capable of quantifying source–target compatibility and dynamically adjusting suppression strength to maximize reusable fine-detail information.

Our ultimate goal is to enable high-performance, training-free model adaptation at minimal computational cost, facilitating deployment of large models in resource-constrained environments and accelerating the development of efficient AI systems on diverse devices.

## G  LLM USAGE STATEMENT

A Large Language Model (Google's Gemini) was used as a general-purpose writing assistant, primarily for improving grammar and rephrasing sentences for clarity. The core scientific ideas, experimental design, results, and their interpretations were conceived and articulated entirely by the human authors. The authors have carefully reviewed, edited, and take full responsibility for all content in this paper, ensuring its scientific accuracy and originality. No part of the research ideation, experimental implementation, or data analysis was performed by an LLM.

