# OpenReview forum: "One-for-All Model Initialization with Frequency-Domain Knowledge"
_ICLR.cc/2026/Conference — Submitted to ICLR 2026_

### Official Review · Reviewer_1adk · 2025-10-19

**Soundness:** 2
**Presentation:** 2
**Contribution:** 2
**Rating:** 2
**Confidence:** 4

**Summary:**

The paper proposes FRONT, a frequency-domain method for initializing neural networks by extracting low-frequency components (via Discrete Cosine Transform, DCT) from pretrained weights. The authors argue that these components—termed “learngenes”—capture task- and architecture-agnostic knowledge, allowing models of various sizes (e.g., different depths or widths) to inherit such knowledge through a training-free process. They also propose FRONT+, which introduces a spectral regularization term to refine these learngenes through a brief fine-tuning process. The authors report experimental performance across: vision models (DeiT, ResNet) and language models (BERT, RoBERTa, GPT2) via various downstream datasets (classification, detection, segmentation, GLUE benchmark). They claim up to 15× faster convergence and 40% less FLOPs compared to training from scratch.

**Strengths:**

1. The use of DCT to extract transferable low-frequency components for cross-architecture initialization is interesting. It provides a fresh perspective on model reuse and transfer learning.
2. The paper offers experiments on both vision and language domains, demonstrating broad applicability and consistent improvements.
3. The proposed method operationalizes the abstract “learngene” concept in a concrete, reproducible way—turning a theoretical notion into a working initialization strategy.

**Weaknesses:**

1. It lacks rigorous theoretical analysis to support why low-frequency components encode general knowledge. The claim that low-frequency weights correspond to “learngenes” remains speculative.
2. Unlike images, there is no inherent spatial ordering of weight indices. Applying DCT assumes a kind of smoothness across indices that is not theoretically justified.
3. The paper doesn't show that low-frequency weights correspond to smoother or more general representations in the activation space.
4. Although the authors test across multiple datasets, the analysis lacks examination of negative cases—when and why the method fails. There's also little discussion on transfer to fundamentally different architectures.

**Questions:**

1. Why DCT rather than Fourier, Wavelet, PCA, or SVD? The authors only cite DCT’s “energy compaction” property from image compression.
2. What is the definition of high/low frequency in weight space?

---

> ### Author Response · Authors · 2025-11-22
>
> Dear Reviewer 1adk,
>
> We sincerely thank you for highlighting our use of DCT for transferable low‑frequency initialization, recognizing its fresh perspective on model reuse, its broad applicability across vision and language, and our concrete operationalization of the “learngene” concept.  Below, we provide our detailed responses to your comments and suggestions. We have also updated our paper and added further sections and points in the paper that address the points/questions raised by the reviewer.
>
>
> > **C1.** It lacks rigorous theoretical analysis to support why low-frequency components encode general knowledge. The claim that low-frequency weights correspond to “learngenes” remains speculative.
>
> Thank you for the insight! We first acknowledge that *learngene lacks a rigorous mathematical definition*. It generally serves as a concept for a carrier of compact, generalizable common knowledge. Prior methods typically instantiated this concept heuristically, and our work builds upon this paradigm. A key distinction of our approach is its superior flexibility, allowing for seamless adaptation across varying depths and widths.
>
> Admittedly, a gap between theoretical justification and empirical practice *remains a common challenge in this field*. Bridging this gap is a crucial direction for future research. We thank the reviewers for this critical comment. To this end, we have provided a theoretical analysis in **Appendix E**, where we humbly attempt to formalize why low-frequency components serve this role effectively.
>
> In a nutshell, building on the established spectral bias phenomenon (Rahaman et al., 2019) which implies pre-training prioritizes smooth functions, we prove that at the start of fine-tuning, the optimization gradients are almost entirely concentrated in the high-frequency subspace to fit the sharp task details. This mathematically confirms why low-frequency weights remain naturally stable and serve as the learngene, while high-frequency weights must change significantly to handle adaptation.
>
>
>
> (Rahaman et al., 2019) On the Spectral Bias of Neural Networks, ICML'19.
>
>
>
>
> > **C2.** Unlike images, there is no inherent spatial ordering of weight indices. Applying DCT assumes a kind of smoothness across indices that is not theoretically justified.
>
> Thank you for raising this point. Many prior works have successfully treated network weights as structured signals, analogous to images, such as generating weights via diffusion models [1,2,3,4], or compressing models using DCT [5,6]. These approaches implicitly leverage the fact that weight tensors possess architectural axes rather than being an unordered set of parameters. Empirical and theoretical studies [7,8,9] show that weights exhibit strong cross‑layer correlations, forming predictable structures along these axes.
>
> While smoother signals yield stronger energy compaction in the frequency domain, the DCT is applicable to any discrete finite‑length signal and **does not require smoothness assumptions [10]**. As noted by [10], *"In pattern recognition, orthogonal transforms enable a noninvertible transformation from the pattern space to a reduced dimensionality feature space... implemented with substantially less features, with only a small increase in classification error."* This highlights that the effectiveness of DCT extends well beyond perfectly smooth signals.
>
> Our experiments in Sec. 4 validate this principle: retaining only low‑frequency coefficients consistently preserves downstream accuracy and accelerates convergence (up to 15×), highlighting its utility as an effective and transferable initialization strategy across architectures.
>
>
>
> [1] Mixue, et al. Weight diffusion for future: Learn to generalize in non-stationary Environments, NeurIPS’24.
>
> [2] Wang et al. Scaling up parameter generation: A recurrent diffusion approach, NeurIPS’25.
>
> [3] Soro et al. Diffusion-based neural network weights generation, ICLR'25.
>
> [4] Wang et al. Neural network diffusion. arxiv preprint arxiv:2402.13144, 2024.
>
> [5] Ulicny et al. Tensor reordering for CNN compression, ICASSP'21.
>
> [6] Ulicny et al. Harmonic convolutional networks based on discrete cosine transform. Pattern Recognition, 2022.
>
> [7] He et al. Deep Residual Learning for Image Recognition, CVPR'16.
>
> [8] Xia et al. Initializing variable-sized vision transformers from learngene with learnable transformation. NeurIPS’24.
>
> [9] Lin et al. Linearly decomposing and recomposing vision transformers for diverse-scale models. NeurIPS’24.
>
> [10] Ahmed et al. Discrete Cosine Transform. IEEE Transactions on Computers, 1974.

---

> > ### Author Response · Authors · 2025-11-22
> >
> > > **C3.** The paper doesn't show that low-frequency weights correspond to smoother or more general representations in the activation space.
> >
> > Thank you for this insight. We clarify that our analysis is entirely within the parameter space, and we make no assumption about activation smoothness. The observed low‑frequency structure in trained weights is an emergent property, not tied to activation‑level behavior. Retaining only low-frequency coefficients (“learngenes”) makes our approach applicable across architectures and tasks, independent of activation smoothness hypotheses. While exploring the activation‑space relationship would be an interesting direction, it is orthogonal to our current goal.
> >
> >
> >
> > > **C4.** Although the authors test across multiple datasets, the analysis lacks examination of negative cases—when and why the method fails. There's also little discussion on transfer to fundamentally different architectures.
> >
> > Thank you for the insight. While our analysis focuses on cross‑dataset and cross‑scale transfer, where the method consistently performs well, we agree that documenting boundary conditions is essential. Preliminary experiments indicate that performance degradation occurs when the architectural gap between source and target is extreme—for example, transferring from a Transformer‑based model to a CNN‑based model—due to the lack of shared structural axes that our DCT‑based learner representation exploits. We have added clarifying discussion and examples in *Appendix F* to make these constraints explicit.
> >
> > ---
> >
> > Regarding transfer to fundamentally different architectures, we sincerely thank you for this inspiring suggestion. It points out a potential advantage of our approach: tackling heteromorphic transfer, a task that all the baselines **cannot perform** due to their strict structural constraints, and have *never done* such experiments. In contrast, FRONT operates on the frequency domain of functional units, acting as a universal interface that bypasses these topological barriers. We appreciate the encouragement to explore this frontier, which demonstrates the unique versatility of our method. As suggested, we have added Appendix D.8 on two heteromorphic settings involving simultaneous architectural and size mismatches.
> >
> > 1. BERT ↔ GPT
> >    We tested bidirectional transfer between BERT-B (Encoder) and GPT-B (Decoder), with L12→L6 downscaling in both directions. For GPT-B → BERT-S, FRONT initialization reaches target loss with **15.4%** less pre-training FLOPs than scratch. For BERT-B → GPT-S, FRONT saves **27.0%** FLOPs and achieves equal or better final loss.
> >
> > 2. Standard → Parallel Attention
> >    We transferred knowledge from a standard attention Transformer (DeiT-B L12) to a Mega-ViT-S L6 (Dehghani et al., 2023), which uses parallel attention blocks, changing both attention composition, network depth, and width. Despite the topological mismatch, FRONT initialization reduces the training cost to reach target accuracy on CIFAR-10 by **33.6%** compared to scratch.
> >
> >
> >
> > (Dehghani et al., 2023) Scaling vision transformers to 22 billion parameters, ICML'23.

---

> > > ### Author Response · Authors · 2025-11-22
> > >
> > > > **C5.** Why DCT rather than Fourier, Wavelet, PCA, or SVD? The authors only cite DCT’s “energy compaction” property from image compression.
> > >
> > > Thanks for raising this question. We have added a detailed analysis in *Appendix D.1*. In summary, our choice of DCT is grounded in signal processing principles and the requirement for cross-model transfer.
> > >
> > > **Theoretical Analysis**
> > >
> > > - DCT vs. DFT (Fourier): DCT produces *real-valued* coefficients, matching neural network weights directly. In contrast, DFT outputs complex numbers, which doubles storage and complicates computation. In addition, DFT assumes signals periodicity, creating high-frequency noise at the edges, while DCT treats boundaries as symmetric, keeping the signal smooth and concentrating information much more efficiently.
> > > - DCT vs. PCA/SVD: DCT uses a fixed, universal basis, meaning low frequency always represents the same global patterns regardless of model size, allowing for seamless resizing. In contrast, PCA/SVD generate *data-dependent bases* specific to a single model instance. The principal components of a source model do *NOT* align semantically with those of a target model, making direct transfer mathematically impossible.
> > > - DCT vs. DWT (Wavelets): Our insight, supported by results, is that the transferable knowledge is primarily a global property of the weight distribution. DCT effectively isolates this global spectral structure, whereas DWT is designed for time-frequency localization to capture local features, which is less critical for the specific goal.
> > >
> > > **Empirical Comparison**
> > >
> > > To validate the analysis above, we conducted ablation studies replacing DCT with DFT, DWT, PCA, and SVD for both depth and width transfers. All experimental settings remain consistent with the configurations in Table 1 and Table 2. Consistent with our analysis, DCT-based FRONT yields superior downstream accuracy.
> > >
> > > Since PCA/SVD bases are tied to dimensions, direct transfer is theoretically impossible. To enable a comparison, we performed dimension truncation. While this forces the operation to execute, truncating the dimensions of singular vectors destroys their orthonormality and geometric alignment, leading to significant performance degradation. DWT outperforms DFT and SVD, and DCT still offers a consistent advantage over the localized representation of DWT, confirming the global nature of the transferable knowledge.
> > >
> > > | Method                                 | Scratch | DCT(FRONT) | DFT  | DWT  | PCA  | SVD  |
> > > | :------------------------------------- | :-----: | :--------: | :--: | :--: | :--: | :--: |
> > > | DeiT-Ti\_L12 $\rightarrow$ DeiT-Ti\_L6 |  40.6   |  **63.3**  | 60.8 | 62.0 | 57.9 | 53.0 |
> > > | DeiT-S\_L6 $\rightarrow$ DeiT-Ti\_L6   |  40.6   |  **56.9**  | 47.9 | 55.4 | 46.3 | 43.8 |
> > >
> > >
> > >
> > >
> > > > **C6.** What is the definition of high/low frequency in weight space?
> > >
> > > Thank you for this comment. In our paper, “high” and “low” frequency in weight space *follow the standard definition* from signal processing via the Discrete Cosine Transform (DCT). The DCT decomposes a weight tensor onto fixed cosine basis functions, where low‑frequency components correspond to coefficients of slowly varying basis functions, and high‑frequency components correspond to coefficients of rapidly varying basis functions. We do not introduce a new definition; the distinction is purely mathematical and inherits the conventional meaning from the DCT framework, as formulated in Equation (1).

---

> > > > ### Comment · Reviewer_1adk · 2025-11-23
> > > >
> > > > Thank you for the response. My central concern remains the justification for applying DCT directly to the weight space. The cited work (Rahaman et al. 2019) primarily addresses Fourier analysis in function space, where such analysis is well-established and mathematically meaningful. Similarly, model compression techniques using DCT [5,6] are typically applied to CNNs, where convolutional filters have a clear spatial interpretation and are tightly connected to the structure of the input image; in that context, applying DCT is reasonable.
> > > >
> > > > In contrast, the current work does not justify why applying DCT to raw weights (e.g., of a Transformer), which lack inherent spatial structure. Unless there exists a principled connection from data space to weight space (e.g., some property of the data is transferred into the parameter domain), or a connection from function space to weight, it is unclear why weight tensors should exhibit continuity or frequency structure that makes DCT appropriate.
> > > >
> > > > To illustrate a failure case: consider transformer attention $x W_q W_k^\top x^\top$. If we apply an arbitrary invertible matrix
> > > > $A$ to $W_q$ and $A^{-1}$ to $W_k$, i.e., $W'_q=W_q A$ the computed attention remains unchanged, but the DCT of $W'_q$ can differ dramatically. Thus, the “frequency” structure in weight space is not invariant to function-preserving transformations.
> > > >
> > > > Although Appendix E assumes “there exists a correspondence between the frequency spectrum of the weights and the Lipschitz smoothness of the function”, this assumption requires further justification. Without a clear explanation of the relationship between the weight space and the function space (or data space), with only the empirical evidence shown in Figure 1, it is not sufficient to motivate me to accept the methodology of directly applying DCT to weight tensors.

---

> > > > > ### Author Response · Authors · 2025-11-24
> > > > >
> > > > > Thank you very much for your response!
> > > > >
> > > > > First, we thank for your careful inspection and counter-example. We fully agree with the mathematical validity of your derivation: applying invertible transformations indeed preserves the function output on the source task while changing the weights. However, we respectfully point out that this observation overlooks a critical distinction: **functional equivalence on the source task does not imply transfer equivalence**. As is well-known, transfer learning relies heavily on reusable inductive biases (e.g., spatial locality) encoded in the weights [1,2]. In your case, the re-parameterized weights preserve the *outputs* on the original task, but they can severely disrupt the *spatial organization* of the parameters. While functionally perfect for the fixed source model and task, such weights lose the structural priors needed to generalize to a new task.
> > > > >
> > > > > Thus, you are absolutely right that the DCT representation will also change dramatically in this case, but **this is exactly what should happen**. When the permutation breaks this spatial structure of the weights, it destroys the inductive bias required for transfer. The fact that DCT fails to extract useful information from scrambled weights aligns perfectly with the reality that these weights have lost their value for transfer. We design an experiment to directly test your counter‑example.  We start from our refined pre‑trained models, then apply the following re‑parameterization to the attention block: $W_q' = W_q \cdot A, \quad W_k' = W_k \cdot A^{-1}$, where \(A\) is an invertible diagonal matrix:$A = \mathrm{diag}(1, -1, 1, -1, \dots).$ This transformation preserves functional outputs on the source task but disrupts the spatial organization of weights.  We then use both the original and transformed models to initialize different layers of a DeiT‑Ti backbone, following the same evaluation protocol as *Table 1* on depth scaling.
> > > > >
> > > > > | Method | After $A, A^{-1}$ | Original |
> > > > > | :----- | :-----------------: | :------: |
> > > > > | L4     |        49.6         |   58.7   |
> > > > > | L6     |        57.3         |   63.4   |
> > > > > | L8     |        60.4         |   65.6   |
> > > > > | L10    |        60.8         |   66.8   |
> > > > > | L12    |        61.3         |   67.5   |
> > > > >
> > > > > We also design an experiment on varying downstream datasets, following the same evaluation protocol as *Table 3*.  These results show that destroying the spatial structure significantly degrades transfer performance.
> > > > >
> > > > > | Method   | After $A, A^{-1}$| Original |
> > > > > | :------- | :-----------------: | :------: |
> > > > > | Flower   |        86.4         |   95.1   |
> > > > > | CUB      |        69.2         |   75.2   |
> > > > > | Cars     |        71.5         |   86.1   |
> > > > > | Cifar10  |        94.3         |   96.6   |
> > > > > | Ciafr100 |        77.1         |   80.8   |
> > > > > | Food     |        78.2         |   83.9   |
> > > > > | iNat.    |        57.3         |   65.4   |
> > > > >
> > > > >
> > > > >
> > > > > Second, our method does NOT operate on arbitrary weights, but on those **produced by SGD** (with weight decay). Extensive literature shows SGD inherently biases optimization towards smooth, structurally simple solutions[3,4]. For example, because natural images have strong spatial patterns, the optimizer naturally pushes the model to learn weights that reflect this structure [5]. This is why we see those clear coherent structure in Figure 6. It is NOT an accident. Our use of DCT is justified because it targets this actual, learned structure that standard training pipelines always produce.
> > > > >
> > > > >
> > > > > Finally, we respectfully submit that empirical generalization is the ultimate test of validity. If the weight space were truly as unstructured or the DCT coefficients merely captured spurious patterns, our method would inevitably fail. Yet, the reality is the opposite: FRONT consistently achieves SOTA performance across different architectures, modalities, and tasks. This widespread consistency serves as a **de facto existence proof**: the structural continuity in weight space is real and usable, even while the precise mathematical mapping between weight structure and functional generalization *remains an open challenge for the entire community*.
> > > > >
> > > > > [1] Yosinski et al. How transferable are features in deep neural networks? NeurIPS'14.
> > > > >
> > > > > [2] Kornblith et al.  Do better ImageNet models transfer better? CVPR'19.
> > > > >
> > > > > [3] Keskar et al. On large‑batch training for deep learning: Generalization gap and sharp minima. ICLR'17.
> > > > >
> > > > > [4] Li et al. Visualizing the loss landscape of neural nets. NeurIPS'18.
> > > > >
> > > > > [5] Ainsworth et al. Git re‑basin: Merging models modulo permutation symmetries. ICLR'23.

---

### Official Review · Reviewer_nvmv · 2025-10-26

**Soundness:** 3
**Presentation:** 3
**Contribution:** 3
**Rating:** 6
**Confidence:** 3

**Summary:**

This paper proposes FRONT, a framework that extracts task-agnostic knowledge from pre-trained models by decomposing weights into the frequency domain via DCT and isolating low-frequency components as "learngenes" for initializing models of different sizes. The key empirical observation (Figure 1) shows that low-frequency components remain stable across different model scales and downstream tasks, while high-frequency components are volatile and task-specific. Two variants are proposed: FRONT for direct zero-cost extraction, and FRONT+ with frequency regularization for refinement. Extensive experiments on vision and language tasks demonstrate substantial improvements.

**Strengths:**

1. The concrete instantiation of learngene as low-frequency components is intuitive and creative, with convincing evidence in Figure 1 demonstrating stability of low-frequency components across models and tasks.

2.  FRONT's zero-cost extraction and flexible padding/truncation mechanism make it substantially more practical than training-based methods like GHN-3 and WAVE.

3. The evaluation spans ViT/ResNet/MLP/CNN architectures, multiple datasets, both vision and language domains, and systematic ablations (Figure 5, Table 11) that strengthen the empirical claims.

**Weaknesses:**

1. The frequency ratio r varies by model size (2.2M/3.2M/13.0M for Ti/S/B in Table 1) without principled justification, suggesting $r$ is model-size dependent. This systematic issue is not explored, and hyperparameters like decay rates $γ_d$ in Eq. 6 lack principled selection guidelines.

2. When comparing with training-based methods (WAVE/TLEG), FRONT+ also requires 150 epochs of training, so these should be evaluated separately from FRONT's direct extraction.

3. In Table 3, FRONT occasionally underperforms LiGO (e.g., Flowers: 92.9 vs 94.2), indicating instability; Tables 4-5 show large improvements on detection/segmentation but lack direct comparison with other initialization methods beyond random initialization.

4. Applying 3D-DCT across layer/input/output dimensions (mixing different semantic meanings) without per-layer processing warrants explanation—why not apply DCT separately to each layer?

5. Evaluation only covers homomorphic scaling (BERT-B→BERT-S) without heteromorphic transfer (e.g., BERT→GPT)

**Questions:**

1. Why do low-frequency components specifically encode task-agnostic knowledge, and why DCT over other transforms like Fourier or wavelets? Figure 1 provides only empirical observation, not principled justification.

2. Why does LiGO fail (Table 1, "/") without explanation, and why is knowledge distillation missing as a baseline in vision tasks despite being used for language tasks?

---

> ### Author Response · Authors · 2025-11-22
>
> Dear Reviewer nvmv,
>
> We sincerely appreciate your valuable feedback and your recognition of our method as intuitive, creative, and practical, as well as your acknowledgment of the broad experimental validation. Your thoughtful and detailed review has been highly beneficial, and we are grateful for the time and effort you dedicated to evaluating our manuscript. Below, we provide our detailed responses to your comments and suggestions. We have also updated our paper and added further sections and points in the paper that address the points/questions raised by the reviewer.
>
>
>
>
> >  **C1.** The frequency ratio r varies by model size (2.2M/3.2M/13.0M for Ti/S/B in Table 1) without principled justification, suggesting is model-size dependent. This systematic issue is not explored, and hyperparameters like decay rates in Eq. 6 lack principled selection guidelines.
>
> Thank you for the comments. We believe there may be a misunderstanding between the ratio `r` (which is a single, fixed hyperparameter) and the resulting **absolute number of parameters** transferred (our "learngene"). The hyperparameter `r` itself does not vary between the Ti, S, and B experiments in Table 1. It is a fixed ratio. However, these experiments use different source models (e.g., an open-access DeiT-Ti_L12, DeiT-S_L12, and DeiT-B_L12). Because these source models are of different sizes, applying the same ratio `r` (e.g., 25%) to each of them naturally results in a *different absolute number* of parameters being transferred (a different "learngene" size). For example, 25% of a small DeiT-Ti_L12 has *fewer absolute parameters* than 25% of a large DeiT-B_L12.
>
> We have incorporated a comprehensive ablation in *Sec 4.4.2*. We conducted a grid search on $r$ and $\gamma_d$ using a DeiT-Ti_L8 source model transferring to DeiT-Ti_L10 on ImageNet-1K (Scratch baseline: 46.8%).
>
> | $\gamma_d \setminus r$ | 0.33  | 0.50  | 0.67  | 0.83  | 1.00  |
> | :--------------------: | :---: | :---: | :---: | :---: | :---: |
> |         Direct         | 60.63 | 62.74 | 64.79 | 65.06 | 66.19 |
> |         0.0625         | 66.61 | 66.64 | 66.65 | 66.58 | 66.60 |
> |         0.1250         | 66.97 | 67.22 | 67.02 | 67.11 | 67.13 |
> |         0.2500         | 67.08 | 67.58 | 68.02 | 67.97 | 67.92 |
> |         0.5000         | 66.74 | 66.92 | 66.00 | 66.72 | 66.94 |
> |         1.0000         | 66.57 | 66.76 | 66.60 | 66.69 | 66.65 |
>
> We can observe that for the "Direct" method (FRONT), there is an efficiency-accuracy trade-off. Accuracy improves steadily as $r$ increases (60.63% $\to$ 66.19%). This indicates that in standard pre-trained models, while "learngene" information is low-frequency dominant, it is not strictly confined to the lowest spectrum. Thus, a larger $r$ is required to capture the tail of the knowledge distribution. The proposed spectral refinement ($\gamma_d > 0$) successfully compresses the knowledge into the lowest frequencies. As shown in the table (e.g., $\gamma_d=0.25$), performance becomes remarkably robust to $r$, maintaining ~67.5% accuracy even when $r$ drops to 0.33 or 0.50.
>
> This ablation reveals guidelines for hyper-parameter selection: For zero-cost extraction (FRONT): If cost-insensitive, we recommend a higher $r$ to prioritize accuracy over parameter compactness. For maximum compression (FRONT+): We recommend applying refinement with $\gamma_d \approx 0.25$. This allows for aggressive truncation (e.g., $r=0.33$), maximizing parameter efficiency without compromising accuracy.

---

> > ### Author Response · Authors · 2025-11-22
> >
> > > **C2.** When comparing with training-based methods (WAVE/TLEG), FRONT+ also requires 150 epochs of training, so these should be evaluated separately from FRONT's direct extraction.
> >
> > Thank you for the comment. We would like to respectfully clarify that this is exactly the structure we used in our original submission. As shown in Tables 1, 2, and 3, we explicitly divided the "Methods" into two distinct groups in the first column: **"Direct"** and  **"Train"**.
> >
> >
> >
> > > **C3.** In Table 3, FRONT occasionally underperforms LiGO (e.g., Flowers: 92.9 vs 94.2), indicating instability; Tables 4-5 show large improvements on detection/segmentation but lack direct comparison with other initialization methods beyond random initialization.
> >
> > We sincerely appreciate the reviewer’s observation of FRONT’s substantial gains in cross‑domain detection and segmentation. Regarding Table 3, while LiGO achieves slightly higher accuracy on *Flowers* (92.9 vs 94.2), we view this as a cost–benefit consideration rather than instability: FRONT requires **zero** additional training steps to extract the learngene, whereas LiGO requires over 100 training steps for mapping. Even with this cost gap, FRONT attains the *highest average accuracy* across all datasets in Table 3, suggesting stable and efficient performance in diverse tasks.
> >
> > For Tables 4–5, we note that most existing approaches (LiGO, TELG, WAVE, etc.) are tailored to Transformer architectures and cannot be directly applied to CNN‑based backbones. To the best of our knowledge, no prior initialization method offers both *cross‑domain and cross‑model* applicability for detection and segmentation. We believe these experiments highlight FRONT’s unique generality, and we agree that exploring extended baseline comparisons in these challenging settings is a valuable direction for future work.
> >
> >
> >
> > > **C4.** Applying 3D-DCT across layer/input/output dimensions (mixing different semantic meanings) without per-layer processing warrants explanation—why not apply DCT separately to each layer?
> >
> > We appreciate this question.
> >
> > Our motivation stems from the interdependent structure of deep network parameters. As shown in works like (He et al., 2016), features evolve gradually along the depth dimension rather than changing independently between layers. This means that adjacent layers share correlation. In response to this observation, 3D-DCT *treats the parameter volume as a whole*, compressing this cross-layer information into low-frequency components. In contrast, applying 2D-DCT combined with layer selection may ignore the correlation between layers. The selection strategy typically lacks a principled basis, and inevitably discards the information contained in the unselected intermediate layers, leading to suboptimal initialization. This rationale is supported by our new ablation study below. We use a pretrained DeiT-S_L12 to initialize a target DeiT-Ti_L6. All experimental settings remain consistent with the configurations in Table 1. All methods are identical except for the depth dimension processing.
> >
> > | Method                     | Top-1 Acc. (%) |
> > | :------------------------- | :------------- |
> > | Scratch                    | 40.63          |
> > | 2D-DCT + first-N selection | 50.42          |
> > | 2D-DCT + uniform selection | 49.06          |
> > | **FRONT (Ours, 3D-DCT)**   | **55.71**      |
> >
> >
> >
> > (He et al., 2016) Deep Residual Learning for Image Recognition, CVPR'16.

---

> > > ### Author Response · Authors · 2025-11-22
> > >
> > > > **C5.** Evaluation only covers homomorphic scaling (BERT-B→BERT-S) without heteromorphic transfer (e.g., BERT→GPT)
> > >
> > > We sincerely thank you for this inspiring suggestion. It points out a potential advantage of our approach: tackling heteromorphic transfer, a task that all the baselines **cannot perform** due to their strict structural constraints, and have *never done* such experiments. In contrast, FRONT operates on the frequency domain of functional units, acting as a universal interface that bypasses these topological barriers. We appreciate the encouragement to explore this frontier, which demonstrates the unique versatility of our method. As suggested, we have added Appendix D.8 on two heteromorphic settings involving simultaneous architectural and size mismatches.
> > >
> > > 1. BERT ↔ GPT
> > >    We tested bidirectional transfer between BERT-B (Encoder) and GPT-B (Decoder), with L12→L6 downscaling in both directions. For GPT-B → BERT-S, FRONT initialization reaches target loss with **15.4%** less pre-training FLOPs than scratch. For BERT-B → GPT-S, FRONT saves **27.0%** FLOPs and achieves equal or better final loss.
> > >
> > > 2. Standard → Parallel Attention
> > >    We transferred knowledge from a standard attention Transformer (DeiT-B L12) to a Mega-ViT-S L6 (Dehghani et al., 2023), which uses parallel attention blocks, changing both attention composition, network depth, and width. Despite the topological mismatch, FRONT initialization reduces the training cost to reach target accuracy on CIFAR-10 by **33.6%** compared to scratch.
> > >
> > >
> > >
> > > (Dehghani et al., 2023) Scaling vision transformers to 22 billion parameters, ICML'23.

---

> > > > ### Author Response · Authors · 2025-11-22
> > > >
> > > > > **C6.** Why do low-frequency components specifically encode task-agnostic knowledge, and why DCT over other transforms like Fourier or wavelets? Figure 1 provides only empirical observation, not principled justification.
> > > >
> > > > Thank you for these comments regarding the design choice of the transform basis and the role of low‑frequency components.
> > > >
> > > > 1. Why do low‑frequency components encode task‑agnostic knowledge?
> > > >
> > > > We first acknowledge that *learngene lacks a rigorous mathematical definition*. It generally serves as a concept for a carrier of compact, generalizable common knowledge. Prior methods typically instantiated this concept heuristically, and our work builds upon this paradigm. A key distinction of our approach is its superior flexibility, allowing for seamless adaptation across varying depths and widths.
> > > >
> > > > Admittedly, a gap between theoretical justification and empirical practice *remains a common challenge in this field*. Bridging this gap is a crucial direction for future research. We thank the reviewers for this critical comment. To this end, we have provided a theoretical analysis in **Appendix E**, where we humbly attempt to formalize why low-frequency components serve this role effectively.
> > > >
> > > > In a nutshell, building on the established spectral bias phenomenon (Rahaman et al., 2019), which implies pre-training prioritizes smooth functions, we prove that at the start of fine-tuning, the optimization gradients are almost entirely concentrated in the high-frequency subspace to fit the sharp task details. This mathematically confirms why low-frequency weights remain naturally stable and serve as the learngene, while high-frequency weights must change significantly to handle adaptation.
> > > >
> > > >
> > > >
> > > > (Rahaman et al., 2019) On the Spectral Bias of Neural Networks, ICML'19.
> > > >
> > > > ---
> > > >
> > > > 2. Why do we choose DCT over alternatives such as DFT, DWT, PCA, and SVD?
> > > >
> > > > Thanks for raising this question. We have added a detailed analysis in Appendix D.1. In summary, our choice of DCT is grounded in signal processing principles and the requirement for cross-model transfer.
> > > >
> > > > **Theoretical Analysis**
> > > >
> > > > - DCT vs. DFT (Fourier): DCT produces *real-valued* coefficients, matching neural network weights directly. In contrast, DFT outputs complex numbers, which doubles storage and complicates computation. In addition, DFT assumes signals periodicity, creating high-frequency noise at the edges, while DCT treats boundaries as symmetric, keeping the signal smooth and concentrating information much more efficiently.
> > > > - DCT vs. PCA/SVD: DCT uses a fixed, universal basis, meaning low frequency always represents the same global patterns regardless of model size, allowing for seamless resizing. In contrast, PCA/SVD generate *data-dependent bases* specific to a single model instance. The principal components of a source model do *NOT* align semantically with those of a target model, making direct transfer mathematically impossible.
> > > > - DCT vs. DWT (Wavelets): Our insight, supported by results, is that the transferable knowledge is primarily a global property of the weight distribution. DCT effectively isolates this global spectral structure, whereas DWT is designed for time-frequency localization to capture local features, which is less critical for the specific goal.
> > > >
> > > > **Empirical Comparison**
> > > >
> > > > To validate the analysis above, we conducted ablation studies replacing DCT with DFT, DWT, PCA, and SVD for both depth and width transfers. All experimental settings remain consistent with the configurations in Table 1 and Table 2. Consistent with our analysis, DCT-based FRONT yields superior downstream accuracy.
> > > >
> > > > Since PCA/SVD bases are tied to dimensions, direct transfer is theoretically impossible. To enable a comparison, we performed dimension truncation. While this forces the operation to execute, truncating the dimensions of singular vectors destroys their orthonormality and geometric alignment, leading to significant performance degradation. DWT outperforms DFT and SVD, and DCT still offers a consistent advantage over the localized representation of DWT, confirming the global nature of the transferable knowledge.
> > > >
> > > > | Method                                 | Scratch | DCT(FRONT) | DFT  | DWT  | PCA  | SVD  |
> > > > | :------------------------------------- | :-----: | :--------: | :--: | :--: | :--: | :--: |
> > > > | DeiT-Ti\_L12 $\rightarrow$ DeiT-Ti\_L6 |  40.6   |  **63.3**  | 60.8 | 62.0 | 57.9 | 53.0 |
> > > > | DeiT-S\_L6 $\rightarrow$ DeiT-Ti\_L6   |  40.6   |  **56.9**  | 47.9 | 55.4 | 46.3 | 43.8 |

---

> > > > > ### Author Response · Authors · 2025-11-22
> > > > >
> > > > > > **C7.** Why does LiGO fail (Table 1, "/") without explanation, and why is knowledge distillation missing as a baseline in vision tasks despite being used for language tasks?
> > > > >
> > > > > We thank the reviewer for highlighting these points about our baselines. For LiGO, the “N/A” entries in Table 1 occur because LiGO’s mapping function is explicitly defined for Small to Large transfers through mapping. For example, the source model is L4 and the target is larger (e.g., L4→L6 or L4→L8). When the source and target have the same size (L4→L4), LiGO’s formulation does not apply, so the result is marked “N/A” by design rather than due to an implementation failure.  Regarding Knowledge Distillation (KD) in vision tasks, we agree it is a relevant baseline. The KD experiment for our vision settings has been included in Figure 3.

---

### Official Review · Reviewer_PMdN · 2025-10-31

**Soundness:** 2
**Presentation:** 2
**Contribution:** 2
**Rating:** 4
**Confidence:** 3

**Summary:**

The paper proposes FRONT, a training-free initializer that treats a pretrained network’s weights in the frequency domain. It applies a 3D-DCT to each weight tensor, keeps only the low-frequency coefficients as the compact “learngene,” and reconstructs target-size weights for new models by simple zero-padding/truncation and IDCT. An optional FRONT+ step lightly fine-tunes a source model with a spectral regularizer to make those low-frequency components even more transferable. Experiments show faster convergence and substantial compute savings on different tasks and models.

**Strengths:**

- The proposed method extracts a low-frequency learngene and uses padding or truncating to initialize a variety of models across ViT and CNN. It generalizes well across different depths and width, with minimal computation needed.

- The proposed method speeds up convergence and cuts compute versus scratch or learned-transform baselines.

**Weaknesses:**

- The motivation behind the design is unclear. Why stacking weights across layers and then conduct 3D DCT, what if do this process on 2D weights and then use some selective process to get the learngene?

- The presentation of the experimental results is not that clear, and the experimental settings are concernable. For instance, in table 1, it’s unclear to see what’s the base model in each block is used for initialization? And the results reported in the way of 10-epoch accuracy is not optimal. It should report the final accuracy with the number of epochs of convergence. I would expect a much faster convergence rate of the proposed method versus trivial initialization.

- Results not much improvement over WAVE in Table 1,2,3. Also, why the convergence rate of the proposed initialization method that uses pre-trained knowledge does not show notable advantages over traditional methods?

- Lack ablation studies on deciding the ratio $r$.

**Questions:**

- What if the architecture is different? For example, the transferring standard attention block to the parallel attention block in [1]?

- There are tons of pre-trained models in the model zoo, any principles to select one as the learngene to initialize future trianing?

- What’s the design choice of using DCT, what about DFT, DWT and other basis?



1.	Dehghani, Mostafa, et al. "Scaling vision transformers to 22 billion parameters." International conference on machine learning. PMLR, 2023.

---

> ### Author Response · Authors · 2025-11-22
>
> Dear Reviewer PMdN,
>
> Thank you for your careful review of our manuscript. We sincerely appreciate your recognition of the cost‑efficient, high‑performance model initialization enabled by our method, as well as your acknowledgment of the comprehensiveness of our evaluation. Below, we provide detailed responses to your comments. We have also updated our paper and added further sections and points in the paper that address the points/questions raised by the reviewer.
>
>
>
> > **C1.** The motivation behind the design is unclear. Why stacking weights across layers and then conduct 3D DCT, what if do this process on 2D weights and then use some selective process to get the learngene?
>
> We appreciate this question.
>
> Our motivation stems from the interdependent structure of deep network parameters. As shown in works like (He et al., 2016), features evolve gradually along the depth dimension rather than changing independenty between layers. This means that adjacent layers share correlation. In response to this observation, 3D-DCT *treats the parameter volume as a whole*, compressing this cross-layer information into low-frequency components. In contrast, applying 2D-DCT combined with layer selection may ignore the correlation between layers. The selection strategy typically lacks a principled basis, and inevitably discards the information contained in the unselected intermediate layers, leading to suboptimal initialization. This rationale is supported by our new ablation study below. We use a pretrained DeiT-S_L12 to initialize a target DeiT-Ti_L6, including simultaneous depth and width scaling. All experimental settings remain consistent with the configurations in Table 1. All methods are identical except for the depth dimension processing.
>
> | Method                     | Top-1 Acc. (%) |
> | :------------------------- | :------------- |
> | Scratch                    | 40.63          |
> | 2D-DCT + first-N selection | 50.42          |
> | 2D-DCT + uniform selection | 49.06          |
> | **FRONT (Ours, 3D-DCT)**   | **55.71**      |
>
>
>
> (He et al., 2016) Deep Residual Learning for Image Recognition, CVPR'16.
>
>
>
> > **C2.** The presentation of the experimental results is not that clear, and the experimental settings are concernable. For instance, in table 1, it’s unclear to see what’s the base model in each block is used for initialization? And the results reported in the way of 10-epoch accuracy is not optimal. It should report the final accuracy with the number of epochs of convergence. I would expect a much faster convergence rate of the proposed method versus trivial initialization.
>
> First, we thank you for helping us improve the presentation! We apologize that, due to page limits, the detailed configurations were originally placed in the Appendix, which have now been moved to Section 4.1 in the main text. To clarify, in the *direct* setting, we use publicly available 12‑layer DeiT‑Ti/S/B models as the source to initialize downstream DeiT‑Ti/S/B models. In the *training* setting, we align with the auxiliary model configurations adopted by TELG and WAVE, similarly using 8‑layer DeiT‑Ti/S/B models as the source.
>
> We report 10-epoch accuracy since short-term adaptation is a common standard for evaluating initialization quality (Fu, et al. 2025). As shown in Figure 9, our method achieves performance in just 10 epochs that is comparable to training from scratch for 150 epochs. In response to the reviewer's request for final convergence, we have added the full 300-epoch training curves in Figure 3. As observed, FRONT establishes a dominant lead immediately and maintains superior accuracy throughout the entire training process compared to baselines. This confirms that our method provides both immediate efficiency and optimal final performance.
>
>
>
> (Fu, et al. 2025) Wave: Weight template for adaptive initialization of variable-sized models. CVPR’25.

---

> > ### Author Response · Authors · 2025-11-22
> >
> > > **C3.** Results not much improvement over WAVE in Table 1,2,3. Also, why the convergence rate of the proposed initialization method that uses pre-trained knowledge does not show notable advantages over traditional methods?
> >
> > Thanks for the comment. However, we respectfully disagree with the assessment that the improvement over WAVE is not much or that the convergence rate shows no notable advantages. In model initialization, advantage should be evaluated across *accuracy, cost, and flexibility*:
> >
> > 1. While WAVE performs competitively, it requires training complex auxiliary networks. In contrast, FRONT is **training-free** and extractable in milliseconds. Even our refined variant, FRONT+, achieves higher final accuracy than WAVE while **transferring significantly fewer parameters** (e.g., ~20% reduction in Table 1). Achieving SOTA performance with significantly reduced cost is a substantial advancement. Furthermore, these gains are **not** random fluctuations. We conducted experiments with five random seeds, and t-tests at a 5% significance level confirmed that the performance gaps are significant.
> > 2. Regarding the convergence rate, while the performance advantages of our method might not appear that large, we emphasize that FRONT reaches competitive or superior accuracy from the very early epochs and sustains this level throughout the entire training schedule, without requiring extra computation.
> > 3. FRONT possesses a capability that no existing baseline can achieve, i.e., heteromorphic transfer, as detailed in our response to *C5*. FRONT has the potential to act as a universal interface, enabling effective initialization even across fundamentally different architectures.
> >
> >
> >
> > > **C4.** Lack ablation studies on deciding the ratio $r$.
> >
> > Thanks for pointing this out. We have incorporated a comprehensive ablation in Sec 4.4.2. We conducted a grid search on $r$ and $\gamma_d$ using a DeiT-Ti_L8 source model transferring to DeiT-Ti_L10 on ImageNet-1K (Scratch baseline: 46.8%).
> >
> > | $\gamma_d \setminus r$ | 0.33  | 0.50  | 0.67  | 0.83  | 1.00  |
> > | :--------------------: | :---: | :---: | :---: | :---: | :---: |
> > |         Direct         | 60.63 | 62.74 | 64.79 | 65.06 | 66.19 |
> > |         0.0625         | 66.61 | 66.64 | 66.65 | 66.58 | 66.60 |
> > |         0.1250         | 66.97 | 67.22 | 67.02 | 67.11 | 67.13 |
> > |         0.2500         | 67.08 | 67.58 | 68.02 | 67.97 | 67.92 |
> > |         0.5000         | 66.74 | 66.92 | 66.00 | 66.72 | 66.94 |
> > |         1.0000         | 66.57 | 66.76 | 66.60 | 66.69 | 66.65 |
> >
> > We can observe that for the "Direct" method (FRONT), there is an efficiency-accuracy trade-off. Accuracy improves steadily as $r$ increases (60.63% $\to$ 66.19%). This indicates that in standard pre-trained models, while "learngene" information is low-frequency dominant, it is not strictly confined to the lowest spectrum. Thus, a larger $r$ is required to capture the tail of the knowledge distribution. The proposed spectral refinement ($\gamma_d > 0$) effectively compresses the knowledge into the lowest frequencies. As shown in the table (e.g., $\gamma_d=0.25$), performance becomes remarkably robust to $r$, maintaining ~67.5% accuracy even when $r$ drops to 0.33 or 0.50.
> >
> > This ablation reveals guidelines for hyper-parameter selection: For zero-cost extraction (FRONT): If cost-insensitive, we recommend a higher $r$ to prioritize accuracy over parameter compactness. For maximum compression (FRONT+): We recommend applying refinement with $\gamma_d \approx 0.25$. This allows for aggressive truncation (e.g., $r=0.33$), maximizing parameter efficiency without compromising accuracy.

---

> > > ### Author Response · Authors · 2025-11-22
> > >
> > > > **C5.** What if the architecture is different? For example, the transferring standard attention block to the parallel attention block in [1]?
> > >
> > > We sincerely thank you for this inspiring suggestion. It points out a potential advantage of our approach: tackling heteromorphic transfer, a task that all the baselines **cannot perform** due to their strict structural constraints, and have *never done* such experiments. In contrast, FRONT operates on the frequency domain of functional units, acting as a universal interface that bypasses these topological barriers. We appreciate the encouragement to explore this frontier, which demonstrates the unique versatility of our method. As suggested, we have added Appendix D.8 on two heteromorphic settings involving simultaneous architectural and size mismatches.
> > >
> > > 1. BERT ↔ GPT
> > >    We tested bidirectional transfer between BERT-B (Encoder) and GPT-B (Decoder), with L12→L6 downscaling in both directions. For GPT-B → BERT-S, FRONT initialization reaches target loss with **15.4%** less pre-training FLOPs than scratch. For BERT-B → GPT-S, FRONT saves **27.0%** FLOPs and achieves equal or better final loss.
> > >
> > > 2. Standard → Parallel Attention
> > >    We transferred knowledge from a standard attention Transformer (DeiT-B L12) to a Mega-ViT-S L6 (Dehghani et al., 2023), which uses parallel attention blocks, changing both attention composition, network depth, and width. Despite the topological mismatch, FRONT initialization reduces the training cost to reach target accuracy on CIFAR-10 by **33.6%** compared to scratch.
> > >
> > >
> > >
> > > (Dehghani et al., 2023) Scaling vision transformers to 22 billion parameters, ICML'23.
> > >
> > >
> > >
> > >
> > > > **C6.** There are tons of pre-trained models in the model zoo, any principles to select one as the learngene to initialize future trianing?
> > >
> > > We thank the reviewer for this question. We have to respectfully clarify that our work aims to solve a specific, resource-constrained problem: "**Given a single pre-trained model**, how can we efficiently extract its foundational knowledge *once* and reuse it universally", rather than select a pre-trained model from a model zoo.
> > > Just as a fine-tuning method is expected to work regardless of whether the user chooses BERT or RoBERTa, FRONT is designed to be a universal extraction tool that functions effectively on whatever source model is provided. Establishing selection principles falls outside the scope of this work.

---

> > > > ### Author Response · Authors · 2025-11-22
> > > >
> > > > > **C7.** What’s the design choice of using DCT, what about DFT, DWT and other basis?
> > > >
> > > > Thanks for raising this question. We have added a detailed analysis in *Appendix D.1*. In summary, our choice of DCT is grounded in signal processing principles and the requirement for cross-model transfer.
> > > >
> > > > **Theoretical Analysis**
> > > >
> > > > - DCT vs. DFT (Fourier): DCT produces *real-valued* coefficients, matching neural network weights directly. In contrast, DFT outputs complex numbers, which doubles storage and complicates computation. In addition, DFT assumes signals periodicity, creating high-frequency noise at the edges, while DCT treats boundaries as symmetric, keeping the signal smooth and concentrating information much more efficiently.
> > > > - DCT vs. PCA/SVD: DCT uses a fixed, universal basis, meaning low frequency always represents the same global patterns regardless of model size, allowing for seamless resizing. In contrast, PCA/SVD generate *data-dependent bases* specific to a single model instance. The principal components of a source model do *NOT* align semantically with those of a target model, making direct transfer mathematically impossible.
> > > > - DCT vs. DWT (Wavelets): Our insight, supported by results, is that the transferable knowledge is primarily a global property of the weight distribution. DCT effectively isolates this global spectral structure, whereas DWT is designed for time-frequency localization to capture local features, which is less critical for the specific goal.
> > > >
> > > > **Empirical Comparison**
> > > >
> > > > To validate the analysis above, we conducted ablation studies replacing DCT with DFT, DWT, PCA, and SVD for both depth and width transfers. All experimental settings remain consistent with the configurations in Table 1 and Table 2. Consistent with our analysis, DCT-based FRONT yields superior downstream accuracy.
> > > >
> > > > Since PCA/SVD bases are tied to dimensions, direct transfer is theoretically impossible. To enable a comparison, we performed dimension truncation. While this forces the operation to execute, truncating the dimensions of singular vectors destroys their orthonormality and geometric alignment, leading to significant performance degradation. DWT outperforms DFT and SVD, and DCT still offers a consistent advantage over the localized representation of DWT, confirming the global nature of the transferable knowledge.
> > > >
> > > > | Method                                 | Scratch | DCT(FRONT) | DFT  | DWT  | PCA  | SVD  |
> > > > | :------------------------------------- | :-----: | :--------: | :--: | :--: | :--: | :--: |
> > > > | DeiT-Ti\_L12 $\rightarrow$ DeiT-Ti\_L6 |  40.6   |  **63.3**  | 60.8 | 62.0 | 57.9 | 53.0 |
> > > > | DeiT-S\_L6 $\rightarrow$ DeiT-Ti\_L6   |  40.6   |  **56.9**  | 47.9 | 55.4 | 46.3 | 43.8 |

---

### Official Review · Reviewer_DrRR · 2025-11-05

**Soundness:** 3
**Presentation:** 3
**Contribution:** 3
**Rating:** 6
**Confidence:** 3

**Summary:**

The paper proposes a training-free learngene paradigm and demonstrates that a model’s fundamental, task-agnostic knowledge is encoded in the low-frequency components of its weights and can be effectively inherited by downstream models. Building on this, it introduces FRONT (Frequency domain Knowledge Transfer), a framework that accelerates model convergence.

**Strengths:**

1.The motivation of the paper is clear, and the writing is generally well-structured.
2.The paper provides evidence that task-agnostic knowledge resides in a model’s low-frequency components—an intuitively plausible and insightful finding. It also instantiates the learngene concept as low-frequency representations that can be readily extracted from the model.
3.The experiments are generally thorough and demonstrate the effectiveness of the proposed method.

**Weaknesses:**

Please refer to the Questions section below.

**Questions:**

1.The paper only provides empirical evidence for the knowledge-carrying role of low-frequency components; it appears to lack theoretical support for the claim that “low-frequency components encode task-agnostic knowledge.”
2.FRONT+ enhances low-frequency knowledge by suppressing high-frequency components. But are high-frequency components entirely without transfer value? For example, between similar tasks (e.g., image classification and fine-grained classification), might high-frequency components carry reusable fine-detail information? It would be helpful to further analyze the potential role of high-frequency components.
3.The adaptation of learngene is implemented solely via “truncation / zero padding,” without considering how architectural differences between the source and target models (e.g., differing numbers of Transformer layers or CNN convolutional kernels) affect knowledge mapping. For instance, when the target model has many more layers than the source model, could zero-padded high-frequency regions introduce invalid information and adversely impact model initialization?

---

> ### Author Response · Authors · 2025-11-22
>
> Dear Reviewer DrRR,
>
> We sincerely appreciate your valuable feedback and the recognition of the innovation and performance demonstrated in our work. Your thoughtful and detailed review has been highly beneficial, and we are grateful for the time and effort you dedicated to evaluating our manuscript. Below, we provide our detailed responses to your comments and suggestions. We have also updated our paper and added further sections and points in the paper that address the points/questions raised by the reviewer.
>
> > **C1.** The paper only provides empirical evidence for the knowledge-carrying role of low-frequency components; it appears to lack theoretical support for the claim that “low-frequency components encode task-agnostic knowledge.”
>
> Thank you for the insight! We first acknowledge that *learngene lacks a rigorous mathematical definition*. It generally serves as a concept for a carrier of compact, generalizable common knowledge. Prior methods typically instantiated this concept heuristically, and our work builds upon this paradigm. A key distinction of our approach is its superior flexibility, allowing for seamless adaptation across varying depths and widths.
>
> Admittedly, a gap between theoretical justification and empirical practice remains a common challenge in this field. Bridging this gap is a crucial direction for future research. To this end, we have provided a theoretical analysis in Appendix E, where we attempt to formalize why low-frequency components serve this role effectively.
>
> In a nutshell, building on the established spectral bias phenomenon (Rahaman et al., 2019), which implies pre-training prioritizes smooth functions, we prove that at the start of fine-tuning, the optimization gradients are almost entirely concentrated in the high-frequency subspace to fit the sharp task details. This mathematically confirms why low-frequency weights remain naturally stable and serve as the learngene, while high-frequency weights must change significantly to handle adaptation.
>
>
>
> (Rahaman et al., 2019) On the Spectral Bias of Neural Networks, ICML'19.
>
>
>
> > **C2.** FRONT+ enhances low-frequency knowledge by suppressing high-frequency components. But are high-frequency components entirely without transfer value? For example, between similar tasks (e.g., image classification and fine-grained classification), might high-frequency components carry reusable fine-detail information? It would be helpful to further analyze the potential role of high-frequency components.
>
> Thank you for the insightful comment! We agree that high-frequency components could also be useful. For example, in the case where the source and target tasks are identical, retaining full information is naturally optimal. However, the core motivation of FRONT is to serve as a task-agnostic initialization, where the **downstream task is unknown** during the extraction phase. In this scenario, we *cannot* determine a priori how much high-frequency (HF) information should be retained. Furthermore, the measurement of task similarity and its relationship with the separation of reusable details is highly *subtle*.
>
> To demonstrate the utility of HF components, we conducted an ablation on the suppression strength $\lambda$ in FRONT+, from ImageNet-1K to diverse tasks, where lower $\lambda$ means retaining more high-frequency information. The results reveal that the utility of HF components is unstable and data-dependent. For instance, Stanford Cars suffers catastrophic degradation when retaining too much HF ($\lambda=1e-5$, 52.20% acc), whereas moderate suppression boosts accuracy to 90.43%. Conversely, Flowers102 benefits monotonically from stronger suppression, peaking at the most aggressive setting ($\lambda=5e-03$). This phenomenon confirms that it requires a sophisticated mechanism to quantify source-target similarity, which is beyond the scope of this work. We have added a discussion acknowledging that adaptive HF transfer is a promising direction for future research in Appendix F.
>
> | Target Dataset | λ=1e-05 |  λ=5e-04  |  λ=1e-03  | λ=2e-03 |  λ=5e-03  |
> | :------------- | :-----: | :-------: | :-------: | :-----: | :-------: |
> | Cifar10        |  94.03  | **96.86** |  *96.60*  |  96.47  |   95.86   |
> | Cifar100       |  73.13  |   76.74   | **77.80** |  76.64  |  *77.27*  |
> | Stanford Cars  |  52.20  |   86.86   | **90.43** | *88.20* |   86.54   |
> | Flower102      |  83.18  |   92.97   |   92.08   | *93.85* | **95.38** |

---

> > ### Author Response · Authors · 2025-11-22
> >
> > > **C3.** The adaptation of learngene is implemented solely via “truncation / zero padding,” without considering how architectural differences between the source and target models (e.g., differing numbers of Transformer layers or CNN convolutional kernels) affect knowledge mapping. For instance, when the target model has many more layers than the source model, could zero-padded high-frequency regions introduce invalid information and adversely impact model initialization?
> >
> > We thank you for raising this critical point. Actually, our choice of zero-padding is physically grounded in signal processing, where zero-padding in the frequency domain is mathematically equivalent to interpolation in the spatial domain (Gonzalez & Woods, 2018). This operation effectively performs ideal low-pass filtering and up-sampling, preserving the original structural information without introducing aliasing artifacts.
> >
> > To empirically validate this, we first compared zero-padding against a random-padding baseline which initializes high-frequency regions with Gaussian noise, while keeping the transferred low-frequency knowledge identical. We used ImageNet-1K pre-trained models (DeiT-Ti and DeiT-S) and transferred knowledge to 7 downstream tasks. We can find that zero-padding is always better than random-padding.
> >
> > | Model & Params | Padding | Flowers  | Food101  |  iNAT19  | Cars196  | Cifar10  | Cifar100 |  CUB200  |
> > | :------------- | :------ | :------: | :------: | :------: | :------: | :------: | :------: | :------: |
> > | DeiT-Ti, 3.0M  | Random  |   84.8   |   83.7   |   65.1   |   74.3   |   95.3   |   77.7   |   62.6   |
> > | DeiT-Ti, 3.0M  | Zero    | **95.1** | **83.9** | **65.4** | **86.1** | **96.6** | **80.8** | **75.2** |
> > | DeiT-S, 11.3M  | Random  |   92.1   |   85.2   |   66.6   |   85.3   |   96.1   |   80.4   |   67.7   |
> > | DeiT-S, 11.3M  | Zero    | **97.2** | **86.1** | **68.1** | **89.4** | **97.4** | **84.0** | **78.2** |
> >
> > To further verify zero-padding works when the gap is large, we transferred weights from DeiT-Tiny with 4, 6, 8, and 10 layers to DeiT-Ti with 12 layers. “Fixed‑Low” retains only a fixed number of low‑frequency coefficients. Accuracy remains consistently in the 62.4–63.4% range, which is ~14+% higher than training from scratch, indicating that zero‑padded high‑frequency regions do not introduce harmful effects. “All Freq” transfers all available coefficients from the source model, resulting in higher performance. This improvement aligns with our ablation on $r$. These results demonstrate zero-padding is both robust and effective for universal initialization.
> >
> > | Method                             | Scratch | L4→L12 | L6→L12 | L8→L12 | L10→L12 |
> > | :--------------------------------- | :-----: | :----: | :----: | :----: | :-----: |
> > | Transferred Params (All Freq) / M  |    0    |  2.2   |  3.1   |  4.0   |   4.9   |
> > | Accuracy (All Freq)                |  48.3   |  62.4  |  65.4  |  68.3  |  70.4   |
> > | Transferred Params (Fixed-Low) / M |    0    |  2.2   |  2.2   |  2.2   |   2.2   |
> > | Accuracy (Fixed-Low)               |  48.3   |  62.4  |  63.3  |  63.4  |  63.1   |
> >
> >
> >
> >
> > (Gonzalez & Woods, 2018) Digital Image Processing. 4th Edition.

---

### Meta-Review · Area_Chair_n4gp · 2025-12-28

**Summary:**

This paper presents FRONT, a training-free parameter initialization method that utilizes the Discrete Cosine Transform (DCT) to extract low-frequency "learngenes" from pre-trained models for transfer to variable-sized downstream architectures. The reviewers recognized the novelty and efficiency of using DCT-based low-frequency components but also raised concerns regarding the theoretical validity of applying DCT to non-spatial weight tensors. On the practical side, reviewers questioned the motivation for 3D-DCT over per-layer processing, the sensitivity of hyperparameters, and the magnitude of performance gains compared to baselines like WAVE. Additionally, reviewers also require for more rigorous evaluations involving heteromorphic architecture transfer to justify the "one-for-all" claim.

**Reviewer Concerns:**

The rebuttal addressed most concerns from reviewers, particularly through new ablations justifying 3D-DCT and demonstrating robust performance in heteromorphic transfers (e.g., BERT to GPT). The authors also clarified the experimental setups and provided convincing convergence curves to refute claims of marginal improvements. However, the theoretical dispute with Reviewer 1adk remains outstanding. The authors provided strong empirical evidence that SGD induces a structure that DCT can exploit, and destroying this structure harms transfer. However, a rigorous theoretical link between weight-space frequency and function-space smoothness is still missing.

**Reviewer Scores:**

Reviewer DrRR (Score 6) and Reviewer nvmv (Score 6) would likely to maintain or raise their scores, as the authors provided the requested analysis, hyperparameter ablations, and new heteromorphic transfer experiments.
Reviewer PMdN (Score 4) may maintain the score or increase it to 5, given that the rebuttal clarifying the experimental confusion and showing convergence advantage over baselines.
Reviewer 1adk (Score 2) participated in the discussion but remained unconvinced by the theoretical justification, which may leads to no change with the score.

---

### Decision · Program_Chairs · 2026-01-26

Reject